# Human immunodeficiency virus type 1 impairs sumoylation

Bilgül Mete[1],*, Emre Pekbilir[2],* , Bilge Nur Bilge[3] , Panagiota Georgiadou[2], Elif Çelik[2] , Tolga Sutlu[2] ,
Fehmi Tabak[1],† , Umut Sahin[2],†

**During infection, the human immunodeficiency virus type 1 (HIV-1) manipulates host cell mechanisms to its advantage, thereby controlling its replication or latency, and evading immune responses. Sumoylation is an essential post-translational modification that controls vital cellular activities including proliferation, stemness, or anti-viral immunity. SUMO peptides oppose pathogen replication and mediate interferon-dependent anti-viral activities. In turn, several viruses and bacteria attack sumoylation to disarm host immune responses. Here, we show that HIV-1 impairs cellular sumoylation and targets the host SUMO E1–activating enzyme. HIV-1 expression in cultured HEK293 cells or in CD4+ Jurkat T lymphocytes diminishes sumoylation by both SUMO paralogs, SUMO1 and SUMO2/3. HIV-1 causes a sharp and specific decline in UBA2 protein levels, a subunit of the heterodimeric SUMO E1 enzyme, which likely serves to reduce the efficiency of global protein sumoylation. Furthermore, HIV-1–infected individuals display a significant reduction in total leukocyte sumoylation that is uncoupled from HIV-induced cytopenia. Because sumoylation is vital for immune function, T-cell expansion and activity, loss of sumoylation during HIV disease may contribute to immune system deterioration in patients.**

## Introduction

The human immunodeficiency virus type 1 (HIV-1) replicates primarily in CD4+ T-helper lymphocytes of the immune system, and to a limited extent in dendritic cells and macrophages (1). Along the course of untreated HIV-1 infection, as the viral load progressively increases, the immune function collapses due to the depletion of circulating CD4+ T-cells (1, 2, 3). Clinical manifestations of AIDS include various opportunistic infections and specific cancers, and often present once the number of CD4+ T-cells falls below 200 cells/mm$^3$ (4). During infection,

HIV-1 interfaces with multiple cellular pathways and proteins, hijacking some to its advantage for efficient replication or for latency control, and targeting others to disarm host immune responses (5, 6, 7, 8). Among these, post-translational modifications (PTMs) constitute an emerging class of interactors that can either promote or restrict viral replication (9, 10, 11, 12). For example, the spike proteins gp120/gp41 are subject to glycosylation before their cleavage from the gp160 precursor (Env), which is necessary for further maturation and assembly (13, 14, 15). Phosphorylation of HIV-1 reverse transcriptase by Cdk2 or casein kinase II (CK2) can stimulate its distinct functions such as RNA-dependent polymerization or ribonuclease H activity (9, 16, 17, 18, 19). HIV-1 integrase is a key drug target that plays a central role in the incorporation of viral DNA into the host genome. This critical enzyme undergoes multiple PTMs including phosphorylation, acetylation, ubiquitination, and sumoylation (20). The biochemical consequences of these modifications are diverse, some modulating the affinity of integrase to DNA, whereas others regulate the enzyme's stability.

Sumoylation is an essential PTM with crucial roles in eukaryotic cells (21). Sumoylation involves reversible modification of substrate proteins on target lysine residues by covalent conjugation of a small, 18 kD, ubiquitin-like peptide called small ubiquitin-like modifier (SUMO) (21, 22). While SUMO shares only 18% sequence identity with ubiquitin, the overall structural similarity between these two peptides is remarkable (23). Similar to the ubiquitylation cascade, SUMO attachment occurs through the coordinated efforts of a set of enzymes, including the E1 (activating) and the E2 (conjugating) enzymes, as well as a limited set of SUMO E3 ligases (21, 22, 24). In the first step of this conjugation cascade, the mature SUMO peptide is activated in an ATP-dependent manner by the SAE1/UBA2 heterodimer (E1 enzyme). After activation, SUMO is transferred onto UBC9, the unique E2 enzyme of the pathway, which is capable of directly catalyzing the attachment of SUMO to target lysine residues that reside in specific sumoylation motifs on substrate proteins. Often, these motifs are comprised of ψKxD/E where ψ denotes a large hydrophobic residue and x any residue,

[1]Department of Infectious Diseases and Clinical Microbiology, Istanbul University-Cerrahpasa, Cerrahpasa School of Medicine, Istanbul, Turkey [2]Department of Molecular Biology and Genetics, Bogazici University, Center for Life Sciences and Technologies, Istanbul, Turkey [3]Department of Medical Biology, Istanbul University-Cerrahpasa, Cerrahpasa School of Medicine, Istanbul, Turkey

Correspondence: umut.sahin@boun.edu.tr; ftabak@iuc.edu.tr
*Bilgül Mete and Emre Pekbilir are co-first authors.
†Fehmi Tabak and Umut Sahin are co-senior authors.

however, extended or even reverse consensus motifs have been reported ([21], [22]).

At the biochemical level, conjugation of SUMO can alter various properties of target proteins, including their enzymatic activity, subcellular localization, stability, or interactor profile ([25]). Consequently, a plethora of cellular functions may be modulated by sumoylation, including gene expression and transcription, signal transduction, cell division and proliferation, apoptosis, senescence, stress response, and stem cell renewal ([26], [27]). The essential roles played in the cell by sumoylation have been documented in knockout studies in which abrogation of UBC9 was shown to cause lethality in yeast and mice, resulting from gross defects in cell division, nuclear architecture, and chromatin integrity ([28], [29], [30]).

The human genome encodes four SUMO paralogs: SUMO1, 2, 3, and 4. SUMO2 and SUMO3 share 97% similarity and are collectively referred to as SUMO2/3 ([21]). SUMO2/3 carries an internal sumoylation consensus motif, as well as other non-consensus sumoylation sites, allowing the formation of long SUMO chains with distinct branching patterns, as also observed with ubiquitin ([31]). SUMO1 lacks such an internal consensus motif and thus usually participates in these chains as a terminator cap. Nevertheless, recent studies indicate that under certain stress conditions SUMO1 can also be sumoylated on non-consensus lysines, hinting at the existence of complex mechanisms regulating the poly-SUMO chain topology ([32], [33]). SUMO-targeted ubiquitin ligases (STUbLs) are a specific class of ubiquitin E3 ligases that preferentially recognize poly-SUMO chains and catalyze the addition of ubiquitin either to the growing SUMO chain or to a neighboring lysine residue on substrate proteins ([34], [35]). This way, poly-SUMO chains can serve as a platform to initiate substrate ubiquitination and proteasomal degradation ([34], [35]). Because sumoylated substrates can undergo both proteolytic and non-proteolytic ubiquitination, signaling through poly-SUMO or hybrid SUMO-ubiquitin chains can control a spectrum of cellular responses, especially under stress conditions ([21]). Whereas a limited pool of unconjugated SUMO1 is maintained in vivo, SUMO2/3 is abundantly expressed and rapidly mobilized to be conjugated to target substrates in response to DNA or protein damaging stress, such as heat shock or oxidation ([36], [37], [38]). In line with this observation, SUMO2/3 was found to be essential for cells to survive through hyperthermic shock ([37]). These and numerous other studies therefore indicate that modification by SUMO1 and SUMO2/3 may have distinct physiological outcomes, the latter particularly emerging as a crucial mediator of cellular and organismal responses to physiological or environmental stress conditions, including infection ([21]).

SUMOs indeed play essential roles in anti-microbial defense and also in the immune system ([21], [39], [40], [41]). In vitro in cultured cells, depletion of SUMO proteins favors pathogen replication, whereas SUMO overexpression opposes replication of viruses and bacteria ([42], [43]). For instance, SUMO proteins are now emerging as important mediators of the interferon response ([43]). Interferon stimulation increases the expression of SUMO proteins in a microRNA-controlled manner, and abrogation of sumoylation blunts interferons' antiviral activities in a number of viral infections in vitro, including HIV-1 ([43]). Sumoylation is also intricately linked to other signaling pathways that are critical for the immune system, such as NF-κB and Toll signaling ([39], [44], [45]). At the physiological level, sumoylation regulates T-cell expansion and function, as well as inflammatory responses ([45]).

Remarkably, during infection, several viral and bacterial proteins down-regulate global sumoylation to disarm host immune responses ([45]). Human Herpes Simplex Virus 1 (HSV1) protein ICP0 is among the best characterized of such proteins. ICP0 is a STUbL that targets a subset of sumoylated host proteins for proteasomal destruction, including those with anti-viral activities such as PML and SP100 proteins ([46], [47], [48], [49]). Similarly, during infection by *Listeria monocytogenes*, the bacterial virulence factor listeriolysin O (LLO) induces proteasome-independent degradation of host UBC9, thereby decreasing sumoylation and dampening the innate immune response ([42]). A number of HIV-1 proteins, including the integrase enzyme, are known to be sumoylated ([10], [20], [50], [51]). However, little is known about the reciprocal interactions between HIV-1 and host sumoylation, particularly when it comes to the potentially subversive effects of the former on the latter. In this study, we demonstrate that HIV-1 expression and replication in cultured HEK293 or HeLa cells, or in CD4[+] Jurkat T lymphocytes can cause a dramatic reduction in host cell sumoylation by targeting the UBA2 subunit of the SUMO E1–activating enzyme. We also show that HIV-1 infection in patients is associated with a global demise in total leukocyte sumoylation. Whereas further research is necessary to elucidate the mechanistic basis of sumoylation loss in leukocytes in vivo, our data indicate that this is uncoupled from CD4[+] cell depletion, but rather directly linked to uncontrolled viral replication.

## Results

### HIV-1 impairs cellular sumoylation in vitro

To simulate HIV-1 infection in cell culture, we transfected HEK293 cells or electroporated Jurkat cells, an immortalized line of human CD4[+] T-cell lymphocytes, with a lentiviral vector (pfNL43-dE-EGFP) encoding the HIV-1 genome whose expression is driven by long terminal repeats. This virus is capable of reverse transcription, integration, replication, and expression of all viral genes (including *gag*, *pol*, *tat*, *rev*, *vif*, *vpr*, *vpu*, and *nef*) encoding the structural, accessory, and regulatory HIV-1 proteins in transfected cells, but incapable of forming viral particles due to the impairment of envelope protein production by an EGFP cassette that is inserted within the *env* gene. We verified the transfection (or electroporation) efficiency by following EGFP expression by Western blot (Fig S1A and B). In addition, in both HEK293 and Jurkat cells, Western blot analyses confirmed the expression of viral integrase and Rev proteins (Fig S1A and B).

Remarkably, 48 h post-transfection with the HIV-1 genome, the level of SUMO1-conjugated proteins, as well as of those conjugated with SUMO2/3 declined significantly in HEK293 cells, compared with the cells transfected with a control vector devoid of HIV-1 genes (Fig 1A). Cellular sumoylation by both SUMO1 and SUMO2/3 continued to diminish sharply over the course of 72 h. Critically, we also observed a significant decrease in both SUMO1- and SUMO2/3-conjugated proteins in HIV-1–expressing Jurkat cells (Fig 1B). Importantly, we did not observe any reduction in cellular sumoylation upon transfection or electroporation of these two cell lines with a vector

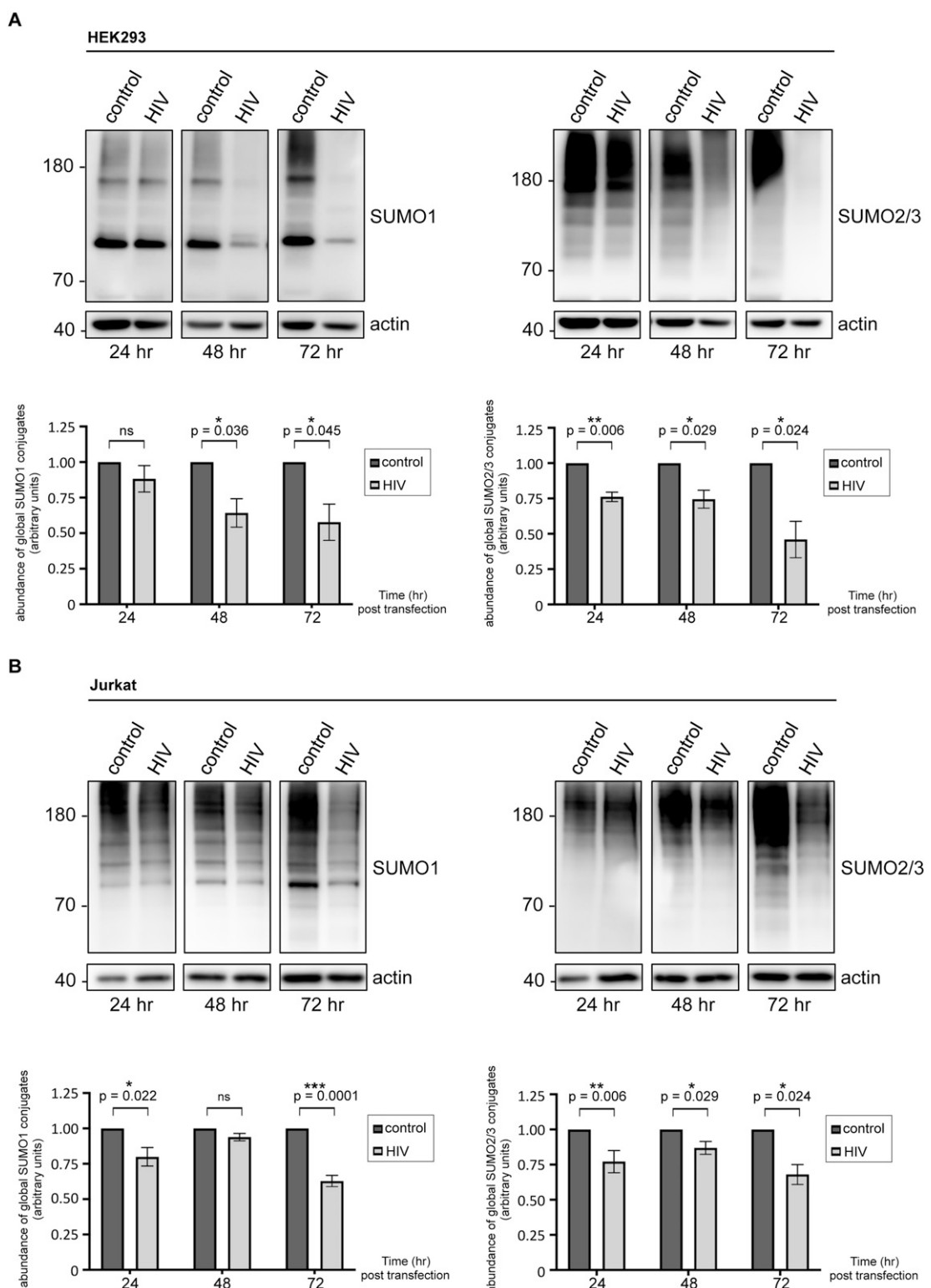

**Figure 1.   HIV-1 diminishes the abundance of cellular SUMO conjugates.**
**(A, B)** A lentiviral vector (pfNL43-dE-EGFP) encoding HIV-1 (containing all nine viral genes: *gag*, *pol*, *tat*, *rev*, *vif*, *vpr*, *vpu*, *nef*, and *env*) was transfected into HEK293 cells (A) or electroporated to CD4[+] Jurkat T lymphocytes (B), allowing reverse transcription and integration, followed by viral protein expression in cells. Neither the envelope protein nor any viral particles are produced because of the insertion of an EGFP cassette within the *env* gene. Expression levels of EGFP and two viral proteins, Rev and integrase, are shown in Fig S1A and B. This system allows in vitro simulation of HIV-1 infection in a safe, reproducible and quantifiable manner. Cells were lysed at indicated times post-transfection, and global sumoylation levels were assessed by Western blot, using human anti-SUMO1 and anti-SUMO2/3 antibodies. Representative

encoding EGFP only (Fig S1C). Similar results were also obtained in HeLa cells expressing the HIV-1 genome (Fig S1D).

Collectively, our data indicate that HIV-1 expression in multiple cell lines, that entails its replication, integration and viral protein production, can precipitate a robust decrease in cell-wide sumoylation, implying that the virus may target this PTM during infection.

### HIV-1 induces a specific loss of UBA2 protein and interferes with SUMOs' conjugation

HIV-1–induced decrease in the abundance of SUMO-conjugated proteins raises a number of possibilities. As mentioned above, poly-SUMOylation can initiate substrate ubiquitination and proteasomal degradation. HIV-1 may induce proteasomal targeting and/or degradation of sumoylated host proteins, resulting in increased turnover. Alternatively, the virus may interfere with the conjugation/deconjugation pathway and antagonize SUMO modification of substrate proteins either by inhibiting the E1, E2, or E3 enzymes or by promoting SUMOs' deconjugation from the modified substrates by enhancing the activities of SUMO-specific proteases (SENPs) (21). To distinguish between these possibilities, we first treated HIV-1–expressing HEK293 cells with MG132, a proteasome inhibitor. As shown in Fig 2A, pharmacologic blockade of the proteasomes did not considerably prevent the virus-induced loss of sumoylated proteins, suggesting that the latter does not reflect enhanced proteasomal turnover. Similarly, in CD4$^+$ Jurkat T-cells, MG132 treatment did not significantly stabilize the cellular SUMO conjugates whose abundance still diminished upon HIV-1 expression (Fig 2B). To further verify proteasome inhibition, we followed the fate of ubiquitin conjugates by Western blot, which accumulated drastically in both cell lines in response to MG132 exposure (Fig S2A and B).

Interestingly, the decline in global high molecular weight SUMO1 conjugates was accompanied by a time-dependent accumulation of the unconjugated SUMO1 peptide, implying that HIV-1 either prevents SUMO1's conjugation to substrate proteins or promotes its deconjugation (Fig 3A). In line with the former possibility, after HIV-1 lentiviral expression in both HEK293 and Jurkat cells, we observed a sharp decrease in UBA2 protein levels, a subunit of the heterodimeric SUMO E1–activating enzyme (Fig 3B). Conversely, SAE1 (the other E1 subunit) and UBC9 (E2 conjugating enzyme) levels remained intact (Fig 3B). Similar results were also obtained in HeLa cells expressing the HIV-1 genome (Fig S1D and data not shown). Contrarily to SUMO1, we did not observe a significant accumulation of the unconjugated SUMO2/3 peptide after HIV-1 expression in HEK293 or Jurkat cells (Fig 3A). This observation likely reflects the well-documented differences in the natural abundance and conjugation behaviour of these two peptides. In cells, free SUMO1 is limiting in quantity, as most of this peptide is found attached to high-affinity substrate proteins, whereas free SUMO2/3 is abundant but remains mostly unconjugated unless cells are exposed to stress

(21, 36). This difference in the conjugation behavior renders SUMO1 highly sensitive and responsive to changes in the enzymatic machinery, and could explain why UBA2 hampering by HIV-1 leads to a dramatic accumulation of this peptide, while not noticeably altering the abundance of free SUMO2/3. However, we cannot exclude the possibility that HIV-1 may also modulate SUMO1 gene expression, while not altering the expression of SUMO2/3.

Subsequently, we attempted to understand the basis of the HIV-1–induced decline in UBA2 expression. To address this point, we first checked the UBA2 mRNA levels after transfection or electroporation of HIV-1 in HEK293 or Jurkat cells. Real-time PCR analyses did not indicate considerable changes in UBA2 transcript levels, suggesting that the virus did not impede the transcription of UBA2 gene (Fig S3A). To test whether HIV-1 enhances the degradation of UBA2 protein by the proteasomes, we used MG132 to inhibit the latter. Nevertheless, proteasomal blockage did not considerably stabilize UBA2 in HIV-1–expressing cells (Fig S3B), hinting at the existence of non-proteasomal proteolytic mechanisms that may consume this protein upon viral entry/expression.

Collectively, these data demonstrate that HIV-1 specifically targets the UBA2 subunit of the host SUMO E1 enzyme and induces its non-proteasomal degradation, thereby reducing the efficiency of global SUMO conjugation.

### Global leukocyte sumoylation is diminished independently of HIV-induced cytopenia during HIV disease

Because sumoylation is critical for immune function and for anti-viral defense, we investigated whether the sumoylation profile of the immune system is affected in vivo in HIV-1–infected individuals. Selective isolation of HIV-1–infected CD4$^+$ T-cells from patients would be challenging, and we wanted to monitor a wider fraction of the immune system. Hence, we collectively analyzed the total leukocyte population, which includes PBMCs and granulocytes.

We collected peripheral blood samples from 19 ART (antiretroviral therapy)-naive HIV-1–infected (HIV(+)) patients, as well as from 12 healthy donors belonging to an uninfected control group (HIV(−)). Patients belonged to a similar age-group with no known comorbidities (no diabetes, hypertension, cancer, thyroid, or cardiovascular disease) or co-infections (except patient #11 who tested positive for HBV: hepatitis B virus [Table 1]). Total leukocytes, including PBMCs and granulocytes, were isolated and subjected to protein extraction, followed by SDS–PAGE and immunoblotting with anti-SUMO1 or anti-SUMO2/3 antibodies to assess global sumoylation levels. In line with our in vitro findings, compared with the HIV(−) group, the HIV(+) group displayed on average a 22% (±0.064) decrease in global leukocyte sumoylation by SUMO1 ($P$ = 0.004), along with a 33% (±0.061) decrease in sumoylation by SUMO2/3 ($P$ < 0.0001) (Fig 4A). These findings suggest that HIV-1 may target global immune system sumoylation in vivo, possibly more severely affecting the SUMO2/3-conjugated proteins. Critically, equal amounts of total protein were loaded on gels when comparing HIV(−) and

---

blots are shown. Densitometric quantifications of SUMO signals were normalized to actin expression (which serves as a loading control). Data are presented as mean ± SEM (n = 4 for HEK293 cells, n = 7 for Jurkat cells), asterisks denote statistical significance ($P$-values were calculated using $t$ test assuming unequal variances, ns, not significant). Control: cells transfected with an empty vector backbone devoid of HIV-1 genes, HIV, cells expressing the HIV-1 genome.

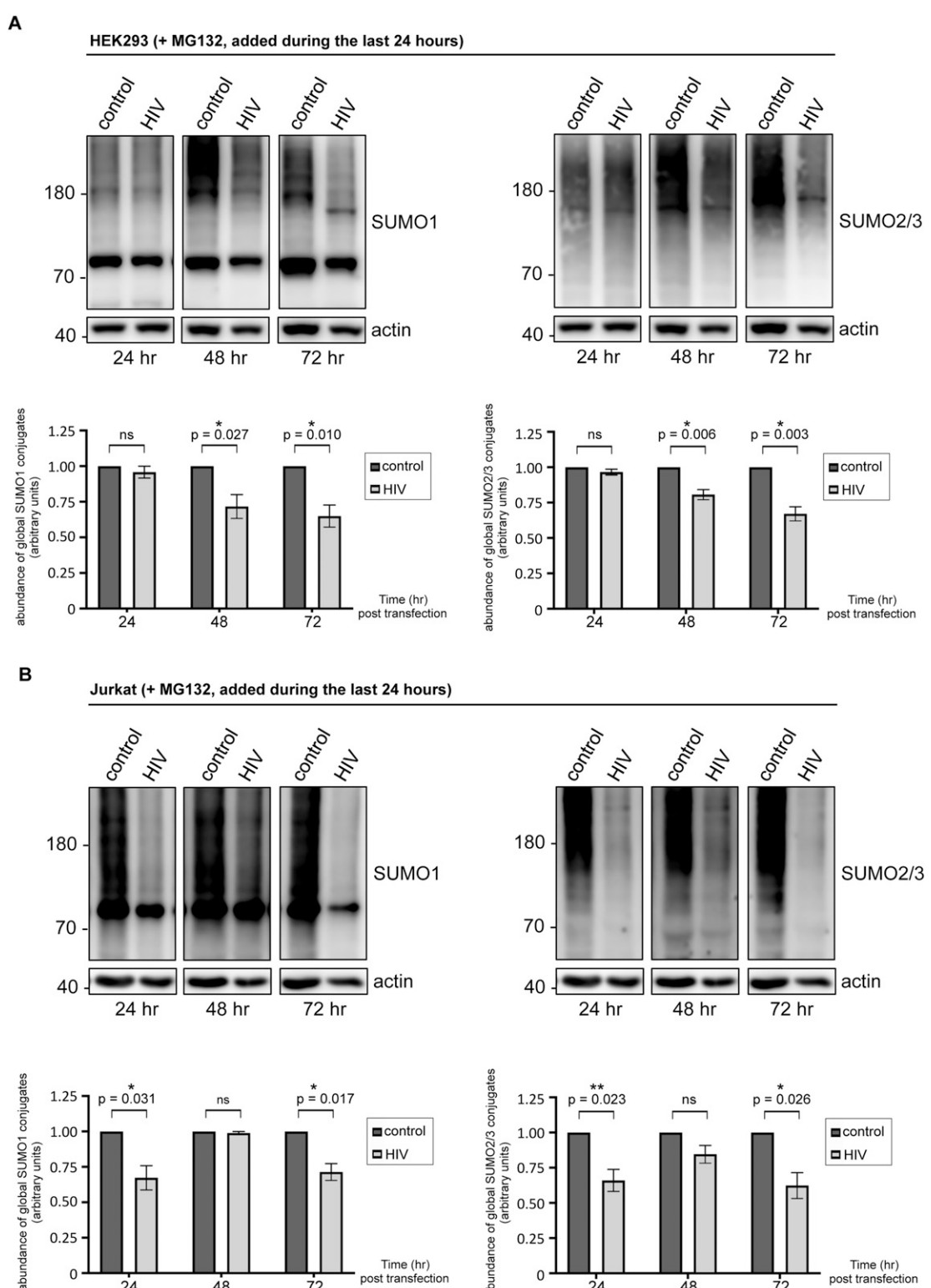

**Figure 2. HIV-1 induces proteasome-independent loss of cellular SUMO conjugates.**
**(A, B)** HEK293 cells (A) or CD4+ Jurkat T lymphocytes (B) were transfected or electroporated with the lentiviral vector pfNL43-dE-EGFP encoding the HIV-1 genome as described above, then treated for 24 h with 3 $\mu$M MG132, a proteasome inhibitor drug, before lysis (at indicated times post-transfection/electroporation). Global SUMO1 and SUMO2/3 conjugates were analyzed by Western blot. Densitometric quantifications of SUMO signals were normalized to actin expression (which serves as a loading control). Data are presented as mean ± SEM (n = 5 for HEK293 cells, n = 4 for Jurkat cells), asterisks denote statistical significance (P-values were calculated using t test assuming unequal variances, ns: not significant). Control: cells transfected with an empty vector backbone devoid of HIV-1 genes, HIV: cells expressing the HIV-1 genome. Ubiquitin blots are shown in Fig S2.

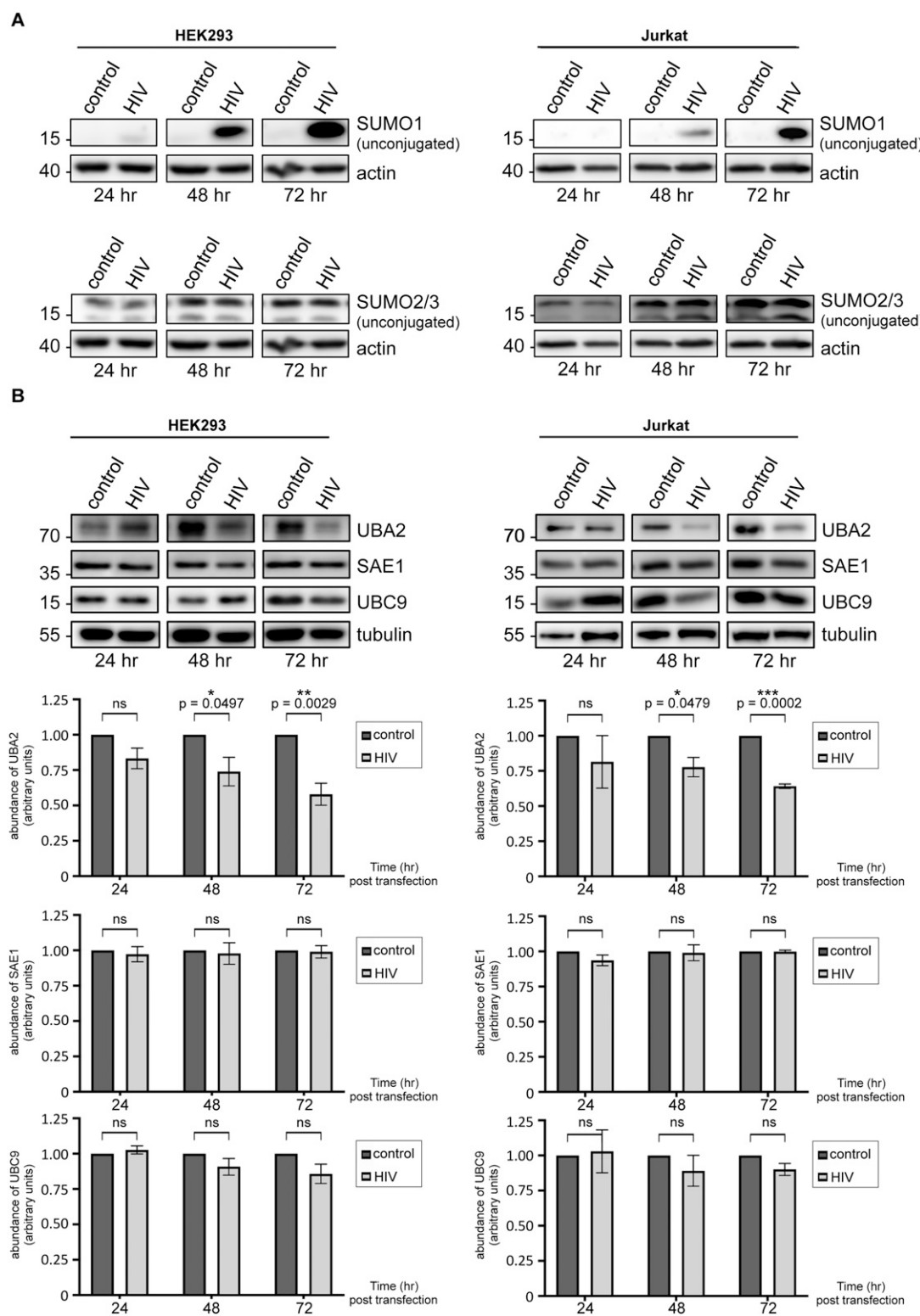

**Figure 3. HIV-1 targets UBA2, a subunit of the SUMO E1–activating enzyme.**

**(A)** HEK293 cells or CD4[+] Jurkat T lymphocytes were transfected or electroporated with the lentiviral vector pfNL43-dE-EGFP encoding HIV-1 as described above, followed by lysis at indicated times post-transfection to assess the level of free (unconjugated) SUMO1 and SUMO2/3 proteins by Western blot. **(B)** Western blots show UBA2, SAE1 and UBC9 protein levels in HEK293 and Jurkat cells. Cells were lysed and proteins were analyzed at indicated times after transfection or electroporation with HIV-1. HIV-1 specifically down-regulates the host UBA2 protein, a subunit of the heterodimeric SUMO E1 enzyme. Densitometric quantifications of UBA2, SAE1, and UBC9 signals were normalized to tubulin expression, which serves as a loading control. Data are presented as mean ± SEM (n ≥ 3 for HEK293 and Jurkat cells), asterisks denote statistical

HIV(+) leukocyte profiles, suggesting that the reduction (or total loss in some cases) in global sumoylation in HIV(+) samples does not simply reflect CD4$^+$ cell (or overall protein) depletion. To confirm this, we artificially depleted the CD4$^+$ cells from the peripheral blood of HIV(−) individuals in vitro (Fig S4A). Western blot analysis of the remaining CD4$^-$ fraction did not indicate a significant decrease in global SUMO conjugate levels (Figs 4B and S4B), implying that the contribution of CD4$^+$ cells to overall leukocyte sumoylation is low. In addition, we studied three patients (patients #3, #4 and #12 in Table 1) who received ART during the course of this study, and on average, displayed a 60.5% (±0.014) decrease in global sumoylation by SUMO2/3 before the initiation of treatment (n = 3, $P$ = 0.0008) (Fig 4C). Whereas all three patients achieved undetectable HIV-1 viral loads as early as 3 mo into the treatment, their CD4$^+$ cell counts did not considerably increase (remained low between 80 and 250/mm$^3$, a normal CD4$^+$ count ranges from 500 to 1,200 cells/mm$^3$); nevertheless, global leukocyte sumoylation levels were restored concomitant with this viral suppression (Fig 4C). As a consequence, these results suggest that the reduction in global leukocyte sumoylation is uncoupled from HIV-induced cytopenia and likely a direct effect of uncontrolled viral replication and HIV-1 interaction with the immune system.

We then asked whether a potential decrease in UBA2 expression in vivo may contribute to the reduced global leukocyte sumoylation activity that we observe in HIV-1–infected individuals. To address this question, we used Western blot analyses to assess UBA2 protein levels in patients' leukocytes. Intriguingly, in line with our findings in cell lines, HIV-1–infected individuals displayed a significant reduction in UBA2 expression (Fig 4D), suggesting that HIV-1 may target UBA2 also in vivo, among others.

All in all, our data indicate that during untreated HIV disease, global leukocyte sumoylation, in particular by SUMO2/3, is down-regulated in the infected patients in a therapy-reversible manner, unraveling another layer of complexity by which HIV-1 targets the host immune system in vivo.

## Discussion

SUMO peptides are critical regulators of innate immunity (21, 45, 52). SUMO is implicated in numerous signaling pathways that regulate immune function, including NF-κB, Toll-like receptor, and interferon signaling. For example, sumoylation finely tunes the processes that oversee interferon production during infection. Several interferon regulatory factors are sumoylated (52). In addition, SUMOs have recently emerged as crucial downstream mediators of interferon anti-viral activities, restricting pathogen replication (43). In general, SUMOs oppose the replication of viruses and bacteria. Overexpression of SUMO1 or SUMO2 in cultured HeLa cells significantly reduces infection by *L. monocytogenes* (42). Conversely, abrogation of sumoylation renders cells vulnerable to infections, for example, UBC9 depletion dramatically increases the efficiency of infection by HSV1 lacking its virulence factor ICP0 (46).

Recent landmark findings indicate that pathogens have evolved with strategies that modulate or impair host sumoylation, or attack sumoylated proteins, to evade host immune defenses (41, 53). As mentioned earlier, ICP0 selectively targets a subset of sumoylated proteins with known anti-viral functions and initiates their proteasomal degradation (46, 47, 49). Epstein–Barr virus encodes microRNAs that silence the expression of proteins, such as PML and RNF4, which play central roles in SUMO signaling (54, 55). Infection by the influenza virus also initiates wide-spread remodeling of the host sumoylation landscape (56). The avian adenovirus CELO encodes a protein called GAM1, which inactivates the SUMO E1 heterodimer and reduces the stabilities of the SUMO E1 and E2 (UBC9) enzymes, thereby efficiently down-regulating host cell sumoylation during infection (57). *L. monocytogenes* globally impairs host sumoylation by initiating non-proteasomal degradation of UBC9 (42). *Shigella flexneri* also abolishes sumoylation during infection by targeting the host UBC9 enzyme (58). A recent study revealed that *Klebsiella pneumoniae* reduces sumoylation to restrict host defense responses by either increasing the levels of SENP2, a SUMO-deconjugating enzyme, or by interfering with let-7 family of microRNAs that target the SUMO transcripts (43, 59). In this study, we provide the first evidence that HIV-1 is also capable of antagonizing cellular sumoylation, most likely via targeting of the host E1 enzyme, and further report that infection in patients is associated with a global demise in total leukocyte sumoylation.

Remarkably, the decrease in global sumoylation in leukocytes was slightly more pronounced for SUMO2/3 than it was for SUMO1 (33% versus 22%, respectively). Growing evidence indicates that protein modification by SUMO1 or SUMO2/3 may lead to different biochemical outcomes and regulate distinct physiological functions (21, 36). For example, in cells exposed to stress conditions such as oxidation and heat shock, SUMO2/3 is rapidly mobilized to be conjugated to thousands of target proteins as part of a major cellular response to cope with environmental assaults. In that respect, both SUMO2 and SUMO3 are essential for cells to survive through hyperthermic stress (37). Because HIV infection inflicts considerable damage and stress on cells, for instance by increasing reactive oxygen species levels (60, 61), the virus may try to cripple host anti-stress mechanisms by primarily disarming SUMO2/3 conjugation. Interestingly, chromatin occupancy by SUMO2/3 may finely tune viral latency or reactivation, or control the expression of host immune-related genes during infection by Kaposi's sarcoma associated herpes virus (62, 63). Of note, we have seen in vitro in HEK293 and Jurkat cells that HIV-1 efficiently reduces sumoylation by both SUMO1 and SUMO2/3, implying that both paralogs may actually antagonize HIV-1 infection in vivo, though further studies are needed to explore this. However, we cannot rule out the possibility that primary and transformed cells may behave differently pertaining to the cellular pathways and mechanisms that are activated or silenced upon viral infection; for instance, paralog-specific suppression of sumoylation.

The virus-induced decline in cellular sumoylation may be explained by the dramatic loss of the UBA2 subunit of the host E1

significance ($P$-values were calculated using $t$ test assuming unequal variances, ns, not significant). Control, cells transfected with an empty vector backbone devoid of HIV-1 genes; HIV, cells expressing the HIV-1 genome.

**Table 1. 19 HIV-1–infected and anti-retroviral therapy (ART)–naive individuals participated in this study.**

| Patient | CD4$^+$ T count | HIV RNA (copy/ml) | Comorbidity and co-infections |
|---|---|---|---|
| 1 | 0 | 62,519 | None |
| 2 | 0 | 281,611 | None |
| 3 | 13 | 506,119 | None |
| 4 | 20 | 909,805 | None |
| 5 | 23 | 1,448,862 | None |
| 6 | 28 | 3,862,507 | None |
| 7 | 30 | 365,300 | None |
| 8 | 32 | 570,578 | None |
| 9 | 39 | 317,400 | None |
| 10[a] | 45 | 5,919 | None |
| 11[a] | 62 | 1,538,498 | Chronic Hepatitis B |
| 12 | 65 | 14,190,707 | None |
| 13 | 91 | 744,873 | None |
| 14[a] | 121 | 935,552 | None |
| 15 | 185 | 216,695 | None |
| 16 | 200 | 3,051,146 | None |
| 17 | 255 | 645,270 | None |
| 18[a] | 343 | 214,343 | None |
| 19 | 350 | 2,717,724 | None |

[a]Only SUMO2/3 data were available for these four patients.
Table shows their CD4$^+$ T-cell counts (in ascending order) and HIV-1 viral loads. Among these, 15 were analyzed for both SUMO1 and SUMO2/3; individuals #10, 11, 14, and 18 could be analyzed for SUMO2/3 only. Except for #11 who had chronic hepatitis B infection, none of the individuals had comorbidities (diabetes, hypertension, cancer, thyroid, or cardiovascular disease) or co-infections. Three of the patients (#3, #4 and #12) later received ART and were further analyzed in Fig 4C.

enzyme. Viral factors targeting UBA2, or signals triggering its non-proteasomal degradation remain to be determined. HIV-1 expression did not cause a remarkable change in UBC9 levels, unlike previously reported for *L. monocytogenes* and *S. flexneri*, which also impair global protein sumoylation (41, 42, 58). Infections often stimulate proteasomal activity resulting in the increased turnover of sumoylated proteins, especially those that are conjugated with SUMO2/3. Indeed, immunoproteasomes are rapidly formed in response to many infections and display enhanced proteolytic activities (64, 65, 66, 67). Alternatively, one of the HIV-1 proteins may have a previously uncharacterized STUbL activity, or may interact with and activate a host STUbL to initiate proteasome-dependent degradation of sumoylated proteins. Importantly, the viral accessory proteins Vif, Vpx (found in HIV-2), and Vpu are known to counteract host restriction factors by serving as adaptors between these factors and certain ubiquitin E3 ligases and inducing their degradation (68). Nevertheless, pharmacologic inhibition of the proteasomes did not prevent or rescue HIV-1–induced down-regulation of host sumoylation, implying that defective conjugation rather than enhanced protein degradation is the most likely mechanism. This is confirmed by the reduction we observed in E1 (UBA2) levels. It remains to be determined whether HIV-1 also interacts with various SENPs to promote SUMOs' deconjugation from protein substrates.

Because of the technical difficulty of isolating and studying patient-derived HIV-1–infected CD4$^+$ cells, we decided to analyze total leukocyte sumoylation in vivo. Remarkably, we observed a significant decline in global leukocyte sumoylation in HIV(+) individuals. Importantly, we showed that the contribution of CD4$^+$ cells, which constituted about 8–15% of all white blood cells (Fig S4A and reference 69), to total leukocyte sumoylation was low, if not negligible (Fig 4B). In addition, ART-induced viral suppression was sufficient to rescue leukocyte sumoylation in vivo, in a manner that was uncoupled from CD4$^+$ cell recovery (Fig 4C). Therefore, we argue that the decrease in global leukocyte sumoylation that we observe in HIV-1–infected individuals does not simply reflect the depletion of CD4$^+$ cells (cytopenia), but is most likely a direct outcome of uncontrolled viral replication and interaction with the host immune system components. Although more research is needed to fully understand the mechanisms leading to HIV-1–associated decline in global leukocyte sumoylation activity, our results suggest that UBA2 loss, at least in part, may contribute to this phenomenon in vivo (Fig 4D). HIV-1 is also known to interact with a subset of non-CD4$^+$ cells (1). The overall effect of this non-canonical interaction may be more severe than previously anticipated, leading to significantly reduced levels of sumoylation in the entire leukocyte population. In addition, HIV-1–induced impairment of CD4$^+$ cell function may disrupt signaling networks that regulate vital activities, such as sumoylation, in other immune cell types collectively (70, 71). Indeed, global cellular sumoylation is responsive to and controlled by major immune regulatory signals,

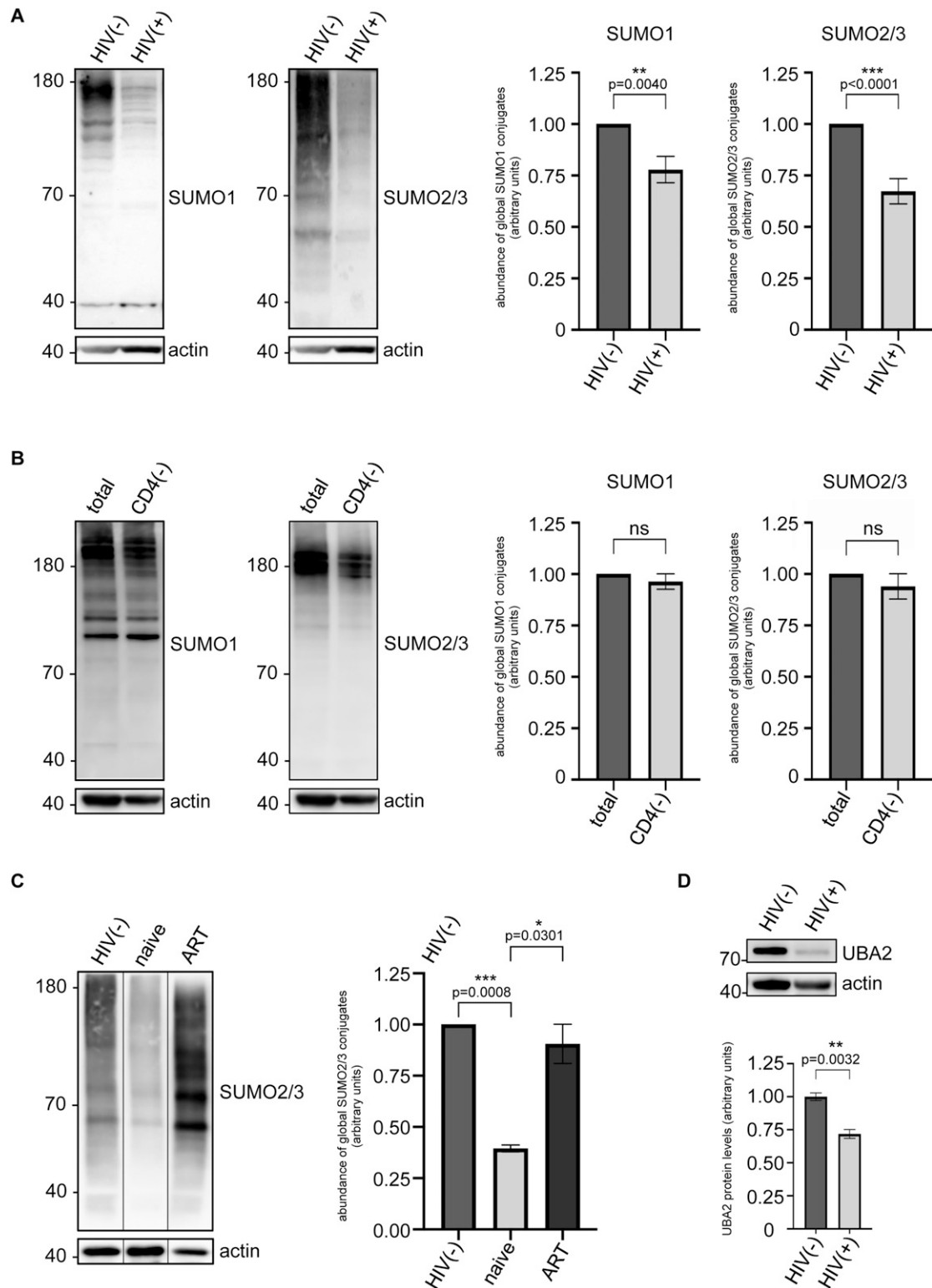

**Figure 4. Global leukocyte sumoylation is impaired during HIV disease.**
**(A)** Western blots show SUMO conjugate levels in total leukocytes of a representative ART-naive HIV-1–infected individual (HIV(+)), in comparison with an uninfected individual (HIV(−)). Western blots were performed as described in Fig 1 and in the Materials and Methods section; sumoylated proteins were detected using human anti-SUMO1 and anti-SUMO2/3 antibodies. Actin: loading control (important note: equal amounts [20 µg] of protein were loaded in each well to allow fair comparison of SUMO profiles between HIV(−) and HIV(+) samples). Graphs (right panel) show densitometric quantifications of SUMO signals normalized to actin expression. Data are presented as mean ± SEM (n = 15 individuals for SUMO1, n = 19 individuals for SUMO2/3), P-values are indicated (using t test assuming unequal variances). Asterisks denote

such as those relayed by Toll-like receptors and interferon receptors, among others (43).

Other AIDS-defining opportunistic pathogens that resurface at low CD4$^+$ counts (<200/mm$^3$) may be potential confounders, possibly contributing to the sumoylation loss in vivo. For example, the CMV, which is known to target leukocytes, is a prominent opportunistic pathogen in advanced HIV disease (72, 73). Of note, only 1 of the 19 patients who participated in this study had a secondary infection other than HIV-1 (patient #11 who tested positive for HBV, Table 1), suggesting that HIV-1 is likely the primary culprit of sumoylation decrease in patients. Clinically successful ART reduces HIV-1 viral load to undetectable levels, thereby restoring immune function, and increases life expectancy. Remarkably, global leukocyte sumoylation was dramatically restored in ART-receiving patients, in a manner that preceded CD4$^+$ cell recovery. The therapy-reversible nature of sumoylation loss in HIV disease is quite intriguing. Considering the critical roles of SUMOs in immune function, the recovery of sumoylation may actually facilitate immune restoration, including CD4$^+$ cell recovery and expansion, in ART-receiving patients.

Interferon α displays limited anti-viral activity on HIV-1, weakly restricting its replication in vitro in cultured cells (43, 74). Interferon's weak anti-HIV-1 activity depends on SUMO proteins and is blunted upon depletion of host SUMOs (43). Because SUMOs are crucial downstream mediators of interferon activity, viral targeting of host sumoylation may explain why interferons remain ineffective in fighting HIV-1 replication in vivo. In line with this hypothesis, although interferons also have a limited impact on HSV1 replication, they strongly restrict the replication of ΔICP0-HSV1, which is incapable of targeting host sumoylation (43, 75). Attacking the UBA2 subunit of the host E1 enzyme seems to be a very effective strategy for HIV-1 to dampen sumoylation. Deletion of HIV-1–encoded virulence factor(s) that attack UBA2 may sensitize the virus to interferon treatment. Importantly, in the clinic, targeting these putative factors by small molecules may be a viable treatment strategy as part of novel combination therapies that may also include interferon α.

HIV-1 disrupts or hijacks multiple cellular mechanisms and pathways upon infection, progressively damaging the immune system at the physiological level. We provide the first evidence that sumoylation is among the cellular processes that are targeted by HIV-1. Given the key functions of sumoylation in innate immunity and anti-viral defense, as well as in cell division, proliferation, and survival, disruption of this vital PTM during HIV-1 infection likely promotes viral replication, and also contributes to immune system deterioration in patients.

# Materials and Methods

## Cell culture, transfections, electroporation, and antibodies

HEK293 and HeLa cells were cultured in DMEM medium (Gibco) supplemented with 10% FCS, and transfected with the lentiviral vector (pfNL43-dE-EGFP) encoding the HIV-1 genome (purchased from Addgene) using the Effectene reagent (QIAGEN), according to manufacturer's instructions. CD4$^+$ Jurkat T lymphocytes were cultured in RPMI-1640 medium (Gibco), and electroporated with HIV-1 using the Neon Transfection System according to manufacturer's instructions. The procedure was performed in Resuspension Buffer R, GFP-positivity was determined using BD Accuri C6 Flow Cytometer.

Human anti-SUMO1 (4930S) and anti-SUMO2/3 (4971S) antibodies were purchased from Cell Signaling Technology and human anti-actin (622102) antibody was purchased from BioLegend. Human anti-tubulin (sc-23948), human anti-GAPDH (sc-32233), human anti-UBA2 (sc-376305), human anti-SAE1 (sc-398080), anti-EGFP (sc-9996), anti–HIV-1-integrase (sc-69721) and anti–HIV-1-Rev (sc-69729) antibodies were purchased from Santa Cruz, and human anti-UBC9 (ab75854) antibody was from Abcam. Human anti-ubiquitin antibody was from R&D Systems (Clone FK2). All antibodies were used at 1:1,000 dilution for Western blot. The proteasome inhibitor drug MG132 was purchased from Sigma-Aldrich and used at 3 μM for 24 h before cell lysis.

## Real-time PCR analyses

The expression level of human UBA2 transcript was determined by quantitative PCR. RNA was isolated from HEK293 or Jurkat cells using Direct-Zol RNA Isolation Kit (Zymogen). cDNA synthesis was performed using SensiFAST cDNA Synthesis Kit (Bioline) according to manufacturer's instructions. RT-PCR was performed using a SensiFAST SYBR No-ROX Kit (Bioline), as well as the following primers targeting UBA2 and GAPDH sequences: UBA2-sense 5′-CCC GAA AGC TAA TAT CGT TGC C-3′; UBA2-antisense 5′-ACT CGG TCA CAC CCT TTT TGA-3′; GAPDH-sense 5′-GGA GCG AGA TCC CTC CAA AAT-3′; GAPDH-antisense 5′-GGC TGT TGT CAT ACT TCT CAT GG-3′ (GAPDH was used as an internal control).

## Ex vivo depletion of CD4$^+$ cells from the peripheral blood

To document the contribution of CD4$^+$ cells to total leukocyte sumoylation, peripheral blood samples were collected from HIV-negative individuals. Isolation of mononuclear and polymorphonuclear leukocytes from

statistical significance. **(B)** The contribution of CD4$^+$ cells to total leukocyte sumoylation was assessed by Western blot after ex vivo depletion of the former from the peripheral blood (leukocytes) of HIV-negative individuals. Total leukocytes (total: before depletion) and the CD4$^+$ cell-deprived fraction (CD4(−)) are shown for comparison for a representative individual. Graphs show densitometric quantifications of SUMO signals normalized to actin expression. Data are presented as mean ± SEM (n = 4 individuals), P-values were calculated using t test assuming unequal variances, ns, not significant. Sumoylation in the CD4$^+$ fraction is shown in Fig S4B. **(C)** Western blot shows leukocyte sumoylation profiles of a representative HIV-1–infected individual before (labeled as naive) and after anti-retroviral therapy (labeled as ART), with respect to that of an uninfected control individual (HIV(−)). Patient received ART for 3 mo after which the HIV-1 viral load became undetectable, yet the CD4$^+$ count did not considerably rise. Graph shows densitometric quantifications of SUMO signals normalized to actin expression. Data are presented as mean ± SEM (n = 3 individuals), P-values are indicated (using t test assuming unequal variances). Asterisks denote statistical significance. **(D)** Western blot analysis of UBA2 in total leukocytes of a representative ART-naive HIV-1–infected individual (HIV(+)), in comparison with an uninfected individual (HIV(−)). 3 HIV(+) and 3 HIV(−) individuals were analyzed; graph shows densitometric quantifications of UBA2 signals normalized to actin expression (data are presented as mean ± SEM, n = 3 individuals, asterisks denote statistical significance; P-value is indicated, using t test assuming unequal variances).

patients' peripheral blood was performed with density-gradient centrifugation using Lymphopure (#426201; BioLegend). After centrifugation, the buffy coat at interphase was collected without disturbing the other fractions and washed twice with PBS. CD4+ cells were depleted using magnetic separation with MS columns on a miniMACS Seperator (Miltenyi Biotec). Briefly, $2 \times 10^6$ PBMCs were labeled with 40 $\mu$l human CD4 Magnetic MicroBeads (#130-045-101; Miltenyi Biotec) in 160 $\mu$l MACS buffer containing PBS (pH 7.2), 0.5% BSA and 2 mM EDTA. Samples were incubated at 4°C for 20 min and washed with 5 ml MACS buffer. The columns were equilibrated using 500 $\mu$l MACS buffer before cells were loaded onto the columns. Unlabeled (CD4−) cells were collected by washing the columns three times with 500 $\mu$l of MACS Buffer. For the collection of labeled cells (CD4+), the columns were removed from the separator and placed on a suitable collection tube and 1 ml of MACS Buffer was run through the columns.

For confirmation of CD4+ depletion, positive and negative fractions were stained with appropriate amounts of anti-human CD3 (#344806; BioLegend), anti-human CD4 (#317415; BioLegend), and anti-human CD8 (#345772; BD BioSciences) antibodies on ice for 30 min. The labeled cells were then washed with PBS and data acquisition was performed on an Accuri C6 (BD BioSciences) instrument. Data were analyzed with FlowJO v10.1 (BD Biosciences) software.

The remaining CD4− fraction (that constituted about 85–92% of total leukocytes, Fig S4A) was subsequently lysed in standard Laemmli buffer containing $\beta$-mercaptoethanol and boiled at 95°C for 10 min. Lysates were cleared upon centrifugation at 11,000$g$ for 10 min. SUMO conjugates were separated on 4–12% gradient gels (Invitrogen) or homemade 8% SDS–PAGE for further analysis by Western blot.

### Patients, leukocyte isolation, protein extraction, and analysis

Patients were selected from those that were admitted to Istanbul University, Cerrahpasa School of Medicine, Department of Clinical Microbiology and Infectious Diseases in Istanbul. Experiments were performed in accordance with the established institutional guidelines and approved protocols from the local ethics committee (Name: Istanbul University-Cerrahpasa, Ethics Committee for Clinical Research, Date and Number: 17 January 2017, 22142), and after obtaining informed consent from all participants. Patients' characteristics are shown in Table 1. Isolation of mononuclear and polymorphonuclear leukocytes from patients' peripheral blood was performed as described above. Isolated leukocytes were subsequently lysed in standard Laemmli buffer containing $\beta$-mercaptoethanol by boiling at 95°C for 10 min. Lysates were cleared upon centrifugation at 11,000$g$ for 10 min. SUMO conjugates were separated on 4–12% gradient gels (Invitrogen) or homemade 8% SDS–PAGE for further analysis by Western blot. Importantly, equal amounts (20 $\mu$g) of total protein were loaded in each well to allow fair comparison of SUMO profiles between different samples, that is, HIV(−) or HIV(+) leukoctye extracts.

### Statistical analyses

Statistical analyses were performed using $t$ test assuming unequal variances, $P$-values were calculated accordingly and indicated on all graphs. All data are presented as mean ± SEM (standard error of mean, n ≥ 3).

## Data Availability

The datasets generated in the current study are available from the corresponding author on reasonable request.

## Ethics Approval and Consent to Participate

Patients were selected from those that were admitted to Istanbul University, Cerrahpasa School of Medicine, Department of Clinical Microbiology and Infectious Diseases in Istanbul. Experiments were performed in accordance with the established institutional guidelines and approved protocols from the local ethics committee (Name: Istanbul University-Cerrahpasa, Ethics Committee for Clinical Research, Date and Number: 17 January 2017, 22142), and after obtaining informed consent from all participants in this study.

## Supplementary Information

## Acknowledgements

The authors thank Arda B Celen (New York University, Grossman School of Medicine) for critically reading the manuscript. This study was funded by Gilead, Inc. (Gilead Turkey Scientific Projects Support to B Mete, F Tabak and U Sahin) and by the European Molecular Biology Organization, Young Investigator Program, Installation Grant (IG3336) to U Sahin.

### Author Contributions

B Mete: formal analysis, investigation, methodology, and writing—original draft.
E Pekbilir: formal analysis, investigation, methodology, and writing—review and editing.
BN Bilge: formal analysis, investigation, and methodology.
P Georgiadou: formal analysis, investigation, and methodology.
E Celik: formal analysis, investigation, and methodology.
T Sutlu: resources, formal analysis, investigation, and methodology.
F Tabak: conceptualization, resources, formal analysis, and writing—original draft, review, and editing.
U Sahin: conceptualization, formal analysis, supervision, funding acquisition, validation, investigation, methodology, project administration, and writing—original draft, review, and editing.

### Conflict of Interest Statement

The authors declare that they have no conflict of interest.

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
