## [Reviewer comments · Life Science Alliance]

Life Science Alliance

Human Immunodeficiency Virus Type 1 Impairs Sumoylation

Umut Sahin, Bilgül Mete, Emre Pekbilir, Bilge Bilge, Panagiota Georgiadou, Elif Çelik, Tolga Sutlu, and Fehmi Tabak
DOI: <https://doi.org/10.26508/lsa.202101103>

Corresponding author(s): Umut Sahin, Bogazici University and Fehmi Tabak, Istanbul University-Cerrahpasa, Cerrahpasa School of Medicine

Review Timeline:

Submission Date:	2021-04-21
Editorial Decision:	2021-07-19
Revision Received:	2022-01-09
Editorial Decision:	2022-02-04
Revision Received:	2022-02-07
Accepted:	2022-02-08

Scientific Editor: Novella Guidi

Transaction Report:

July 19, 2021

Re: Life Science Alliance manuscript #LSA-2021-01103

Prof. Umut Sahin
Bogazici University
Department of Molecular Biology and Genetics
Center for Life Sciences and Technologies
Istanbul 34342
Turkey

Dear Dr. Sahin,

Thank you for submitting your manuscript entitled "Human Immunodeficiency Virus Impairs Sumoylation" to Life Science Alliance. The manuscript was assessed by expert reviewers, whose comments are appended to this letter. All three reviewers raised similar concerns about quantifications, statistics and missing controls to add to the manuscript. We, thus, encourage you to submit a revised version of the manuscript back to LSA that responds to all of the reviewers' points.

Thank you for this interesting contribution to Life Science Alliance. We are looking forward to receiving your revised manuscript.

Sincerely,

B. MANUSCRIPT ORGANIZATION AND FORMATTING:

Reviewer #1 (Comments to the Authors (Required)):

Mete and colleagues reported that HIV diminishes SUMO1 and SUMO2/3 conjugation in both HEK293 and CD4+ Jurkat cells using transfection of HIV vector. Moreover, HIV interferes the conjugation of SUMO1 and SUMO2/3 via inducing a proteasome-independent loss of UBA2 protein in these transfected cells. Consistent with in vitro model, global sumoylation is decreased in leukocytes with HIV patients. Overall, this study was based on physiologically irrelevant HIV transfection system and most of data lacked of direct evidence, making the conclusion was questionable. Several concerns need to be addressed.

1. The authors should include a positive control in the Western blot which could indicate the successful transfection of HIV genome DNA.
2. The level of internal control (actin) in Fig. 1A suggested that the protein loading was not equal.
3. If UBA2 is decreased due to HIV transfection, it is very puzzling that a heavily sumoylated band (the authors claimed HIV integrase) was observed. How come integrase sumoylation was not affected? If the band was not integrase, the result of this heavily sumoylated band did not support the conclusion that decreased UBA2 causes a reduction of global sumoylation. One possible scenario is the heavily sumoylated band competed for cellular protein sumoylation, leading to a decrease of global sumoylation. The author should clarify the issue including the data of integrase sumoylation deficient mutant for the experiments. In addition, how come this heavily sumoylated band was not shown in HIV transfected Jurkat cells or HIV infected patient leukocytes?
4. Is UBA2 protein level also decreased in HIV(+) total leukocytes?
5. Giving that HIV infects CD4+ T cells and completes its replication cycle in approximately 24 hours, the authors should elucidate whether decreased levels of Ubc9 and global sumoylation at 48 hours post-transfection are associated with HIV life cycle.

Reviewer #2 (Comments to the Authors (Required)):

In this manuscript authors explored the SUMOylation events during HIV infection. They observed for the first time that SUMOylation is globally downregulated during HIV infection and that both SUMO1 and SUMO2/3 are involved, in a time dependent manner. Moreover, it seems that UBA2 expression is reduced by HIV, increasing unconjugated SUMO1. Authors showed for the first time that HIV infection induces the downregulation of SUMO pathway. The first results of the manuscript show that HIV impairs cellular SUMOylation in vitro. Data are strongly supportive since there is a significant downregulation of both SUMO1 and SUMO2/3 -conjugated proteins after 24, 48 and 72 hours. In the second figure, authors try to understand the mechanism by which SUMOylation is impaired by HIV. To this end they used the proteasome inhibitor MG132 to exclude a proteasomal degradation of SUMOylated proteins. However, they did not include the quantification of proteins of SUMOylated proteins, nor a control of the effectiveness of the treatment (Ubiquitinated proteins), please include both of them. They then show an increase of unconjugated SUMO1, but not of SUMO 2/3, please include it. Why did authors perform this experiment only in HEK293 cells and not also in Jurkat? Please repeat the experiments in Jurkat cells. The last experiments of the figure show that UBA2 is downregulated by HIV infection, while SAE1 and Ubc9 are not affected and these are strongly supportive data. Authors showed that global leukocytes SUMOylation is impaired independently of HIV-induced cytopenia during HIV disease. To do this they first checked SUMO conjugation levels in HIV+ patients, that seems to confirm the data obtained in vitro and they are strongly supportive. Moreover, they showed that depletion of CD4+ cells from peripheral blood of HIV- individuals has not any effect on SUMO conjugation. The last experiment shows that antiretroviral therapy (ART) is able to rescue the downregulation of SUMO pathway induced by HIV infection and data support the hypothesis. All these last data are strongly supportive.

I have some points in the text that should be addressed:

Authors show in figure 1A, in a SUMO1 western blot a band they claim to be a SUMO1-modified HIV-1 integrase. How do they explain the absence of this band in the same experiment performed in Jurkat cells?

ART acronym is used in the text but not mentioned, please explain it.

In the discussion (page 9), authors mention the effects of viruses on SUMO pathway. It would be complete to also add an older and important finding 10.1016/j.molcel.2004.11.007

Boggio et al Mol Cell 2004

In material and methods authors explain the aim of the experiment. Is it the right section for this?

Minor grammar and spelling errors are present, please check.

Why authors split blot lanes, in particular Figure 3C? Is it possible to show the original blots?

Reviewer #3 (Comments to the Authors (Required)):

In this paper the authors mainly observed that HIV transfected plasmids expression on 293FT and Jurkat cells induce a decrease of sumoylation with a specific decline of UBA2. This study also demonstrates a significant decrease in total leukocyte sumoylation profile from HIV-induced cytopenia, absent from patients under ART treatment. The results seem really promising and the understanding of the mechanism should be a plus for this story, but it seems that it's not the main objective for this publication.

Major points:

Figure 1/S1 and Main text:

Figure S1 shows GFP expression on HEK293 and demonstrated that the HIV plasmid expresses normally the HIV proteins. It should be great to precise the time point of this IF acquisition.

Due to the fact, the authors want to demonstrate a correlation of the decrease of the sumoylation profile (SUMO1 and SUMO2) and the expression and presence of HIV proteins, it should be great to see the GFP profile and another viral protein (as CA expression) on the WB.

The data results from the transfection of an HIV-plasmid on HEK293 or Jurkat cells. In order to be sure that it's not just the over-expression of some proteins that can induce the decline of the sumoylation profile, it should be great to have an internal control with another plasmid that expresses only GFP or another protein.

It should be also interesting to reproduce this data not in the context of transfected cells with a plasmid but with the transduction/infection of the cell lines with the same virus (produced and harvested from HEK293 but pseudotyped with VSV-G for example).

This observation is still observable with an HIV-2 plasmids/viruses?

The authors do the hypothesis that the bands at around 50 to 60 kDa indicated by an arrow (Fig 1A) should be the integrase. I think is pure speculation here. We know that the integrase should be SUMOylated in order to optimize the nuclear localization and the function, but the authors need to prove at least by an anti-Integrase and to observe or not if the WB size are the same. Strangely, during the kinetic we didn't observe any increase of expression of this band (always same intensity @24-48 and 72h post transfection). Moreover, we didn't observe this band on Jurkat cells WB and Fig2A with MG132 treatment exactly the same bands appear on the control line.

It should be a plus to better describe how the abundance profile of SUMO1 and SUMO2/3 is calculated. Is it a quantification performed on a specific bands/molecular weight or on all the profile line?

Figure 2 and Main text:

Fig2C, it should be more significant to be performed and add the quantification of the abundance of UBA2, SAE1 and UBC9 during the kinetic of expression, and also performed to the statistics. We can see that the data are strong but the quantifications. GFP or Capsid expression on the WB should be a plus (but minor here).

Figure 3 and Main text:

It should be interesting to investigate the degradation/decline of UBA2, and also of SAE1 and UBC9 (due to the fact that SUMO2/3 is more impacted than SUMO1, the phenotype of decline if existing can be different).

The authors claimed that the decline of sumoylation observed on SUMO1 and more SUMO2/3 potentially came from HIV-induced cytopenia, absent from patients under ART treatment. Is it possible to put in culture for 24h or 48h PBMCs depleted or not of CD4 cells (healthy donor) and observe or not if the sumoylation profile is changed?

In order to demonstrate that the sumoylation phenotype from total leukocyte came from mainly of the non CD4+ cells, the authors performed a depletion of CD4+ cells from PBMCs. The WB from Fig.3B, show the absence of difference between the total and CD4+ populations.

I understand that the CD4+ cells represent only around 14% of the leukocytes, but it should be great to show the sumoylation profile of the CD4+ purified cells vs. total and CD4+ populations.

Discussion:

it's writing "HIV is also known to interact with a subset of non-CD4+ cells, including macrophages and others [1]." I hope it's just a little issue of writing, but macrophage are CD4+ cells, like dendritic cells and can be infected by HIV, sure less compare to T CD4+ cells.

Minor points:

Figure 1/S1 and Main text:

The HIV plasmid used for this experiment is pNL43-dEEGFP, it should be great to really precise that all the HIV proteins were present unless Env (GFP in place). With this information we know that all accessory proteins (Vpr, Vif, Vpu ...) are present and some of them should participate to this decline of sumoylation. In order to investigate the mechanism (future studies I hope) should be a plus to have pNL43-dEEGFP deleted from Vpr, Vpu, Vif or to over-express only HIV-1 and/or HIV-2 accessory proteins (Vpr, Vpu, Vif, Nef, Vpx ...).

FigS1, that is the profile on HeLa cells of SUMO1 expression after HIV expression? How many times this experiment was performed (n=?)?

Figure 2 and Main text:

The authors observed more SUMO1 (unconjugated) after HIV expression Fig.2B. Is it the case for SUMO 2/3?

Figure 3 and Main text:

Fig 3A, have you try to see if you observed an increase of SUMO1 (unconjugated) on the WB profile (minor)?

Discussion:

it's writing "Of note, we have seen in vitro that HIV efficiently reduces sumoylation by both SUMO1 and SUMO2/3 needed to explore this."

The authors just need to comment/add that for the total leukocyte the sumoylation profile from HIV patient is more impacted on SUMO2/3 than SUMO1. This difference can be easily explain by the difference between immune primary cells and cancer cell lines (HEK293 and Jurkat) that have evolved to cancelled and/or more activate some specific pathway.

it's writing "Alternatively, one of the HIV proteins may have a previously uncharacterized STUbL activity or may interact and activate a host STUbL to initiate proteasome-dependent degradation of sumoylated proteins." The authors should just mentioned that some HIV accessory proteins are specialized to link the E3 ligase complex like Vpr, Vpx in order to induce some proteins degradation.

it's writing "A recent study revealed that *Klebsiella pneumoniae* reduces sumoylation to restrict host defense responses by either increasing the levels of SENP2, a SUMO- deconjugating enzyme, or by interfering with let-7 family of microRNAs that target the SUMO transcripts [43, 58]." Another paper (2014) also demonstrate that interferon controls SUMO availability via the Lin28 and let-7 axis to impede virus replication (DOI: 10.1038/ncomms5187).

Reviewer #1 (Comments to the Authors (Required)):

Mete and colleagues reported that HIV diminishes SUMO1 and SUMO2/3 conjugation in both HEK293 and CD4+ Jurkat cells using transfection of HIV vector. Moreover, HIV interferes the conjugation of SUMO1 and SUMO2/3 via inducing a proteasome-independent loss of UBA2 protein in these transfected cells. Consistent with in vitro model, global sumoylation is decreased in leukocytes with HIV patients. Overall, this study was based on physiologically irrelevant HIV transfection system and most of data lacked of direct evidence, making the conclusion was questionable. Several concerns need to be addressed.

We thank the reviewer for his/her constructive criticism. In particular, we agree with his/her comment that we were unable to employ an in vitro model of infection in our study, which would indeed be valuable and more physiologically relevant. However, setting up such a system that would make it possible to infect cells with live HIV-1 viral particles would require specialized facilities with proper biosafety measures, which we unfortunately lack at our institutions. Therefore, following discussions with the editorial team and upon their approval, we continued our studies with the transfection/electroporation method, and would like to ask for the reviewer's understanding in this matter.

Please note that in this current set up that we use, the HIV-1 lentivirus that we transfect into HEK293 cells or electroporate into CD4+ Jurkat T-lymphocytes is still capable of reverse transcription, integration and replication, as it carries and expresses all of the structural, accessory and regulatory HIV-1 genes: gag, pol, tat, rev, vif, vpr, vpu, nef and env (an EGFP cassette was inserted within env). Therefore, this system can exquisitely simulate viral infection, making it possible to investigate and document potentially detrimental and subversive effects of viral proteins on host cell mechanisms, such as sumoylation.

From this perspective, our study is a first to demonstrate that the introduction of HIV-1 into cells, allowing replication and viral protein expression, can antagonize this critical post-translational modification. We hope that this work will pave the way for further future studies and prompt other groups (with access to proper biosafety measures) to further investigate HIV-1 / host sumoylation interactions in more natural set ups.

In the meantime, we have now extensively modified the manuscript and further enhanced the quality of our data, performed more robust statistical analyses, and

presented additional supportive experimental data, where applicable. For instance, to test whether proteasomal inhibition would prevent the virus-induced decline in cellular SUMO conjugates, we have now also treated the Jurkat cells, alongside HEK293 cells, with MG132 (new Figure 2), and included proper quantifications, statistical analyses and controls. In addition, we now also show data from Jurkat cells to demonstrate the accumulation of the free SUMO1 peptide (new Figure 3A). Furthermore, we quantified the data from new Figure 3B and provided statistical analyses. Importantly, in both HEK293 and Jurkat cells, we verified the expression of viral proteins (upon transfection or electroporation with HIV-1) by following integrase, Rev and EGFP expression in Western blot (Supplementary Figures 1A and 1B). To better understand the mechanistic basis of HIV-1-induced decline in UBA2 expression, we analyzed UBA2 mRNA levels in HIV-1-expressing cells, and assessed whether this phenomenon actually reflected the proteasomal degradation of this protein (Supplementary Figures 3A and 3B).

Please also note that in agreement with the editorial team, to ensure accuracy and correctness, we have decided to use the term "HIV-1" instead of "HIV" throughout the manuscript, including the title.

1. The authors should include a positive control in the Western blot which could indicate the successful transfection of HIV genome DNA.

We thank the reviewer for this important comment. In the original version of the manuscript, we had included immunofluorescence images with positive EGFP signals in the transfected HEK293 cells (note that an EGFP cassette was inserted within the env gene of HIV-1). In the revised manuscript, we now replace this image with much better Western blot data: we now show not only the expression of EGFP, but also of two viral proteins, integrase and Rev, in both transfected HEK293 and electroporated Jurkat cells.

2. The level of internal control (actin) in Fig. 1A suggested that the protein loading was not equal.

The reviewer is correct in that in this representative image of HIV-1 transfected HEK293 cells, there is slightly less actin at 72 hrs (for SUMO1 blot) and at 48/72 hrs (for SUMO2/3). However, this slight difference in loading does not explain the massive decrease in the abundance of global SUMO conjugates at these time points. We performed this experiment many times in HEK293 cells. Indeed, in each experiment, quantifications of global SUMO conjugates, when normalized to actin expression, consistently revealed a dramatic decrease in sumoylation (in a statistically-significant manner when all the data were pooled together, as indicated on the graphs).

3. If UBA2 is decreased due to HIV transfection, it is very puzzling that a heavily sumoylated band (the authors claimed HIV integrase) was observed. How come integrase sumoylation was not affected?

The dynamics of SUMO conjugation/deconjugation differ greatly between specific substrate proteins. Sumoylation of some proteins, especially by SUMO1, is very sensitive and responsive to changes in the SUMO enzymatic machinery (and also to changes in the expression of the SUMO peptides), while sumoylation of others may be more static and slow-responding. For instance, as shown in the figure below (taken from He et al, 2017, Nature Chemical Biology, Probing the roles of sumoylation in cancer cell biology by using a selective SAE inhibitor), pharmacologic inhibition of the SUMO E1 enzyme (by a small molecule inhibitor, ML-792) can efficiently abrogate TRIM28 sumoylation, but not RanGAP1 sumoylation at the same concentration. We have limited, if any, information about the dynamics of HIV-1 integrase sumoylation/desumoylation. It is possible that the sumoylation of this enzyme may be affected by UBA2 loss to a lesser extent than the sumoylation of other cellular proteins.

If the band was not integrase, the result of this heavily sumoylated band did not support the conclusion that decreased UBA2 causes a reduction of global sumoylation. One possible scenario is the heavily sumoylated band competed for cellular protein sumoylation, leading to a decrease of global sumoylation. The author should clarify the issue including the data of integrase sumoylation deficient mutant for the experiments. In addition, how come this heavily sumoylated band was not shown in HIV transfected Jurkat cells or HIV infected patient leukocytes?

Even if this band is not integrase, the possibility exists that it may be a protein whose sumoylation does not respond as quickly to changes in UBA2 levels as other proteins. As the reviewer points out, it is also possible that this heavily sumoylated protein, whether it is integrase or not, may compete for cellular sumoylation. However, we think that this is unlikely for two reasons:

1) *It seems that this protein is not sumoylated by SUMO2/3 following HIV-1 transfection, yet global SUMO2/3 conjugation is still diminished; therefore, it is unlikely that it would compete for SUMO2/3 modification*

2) HIV-1-induced UBA2 depletion likely explains the decrease in cellular sumoylation by both SUMO1 and SUMO2/3.

Because this heavily sumoylated band appears only in HIV-1-infected cells and also because its size corresponds to the size of SUMO1-conjugated HIV-1 integrase, we initially thought and speculated that it could possibly be the sumoylated form of this enzyme. However, upon further consultation with the editorial team, we have come to an agreement that further studies along this line, including determining the identity of this band may be beyond the scope and purpose of this study. We admit that evaluating whether a virus with 'sumoylation-deficient integrase' would also cause a reduction in cellular sumoylation would be very interesting, however these studies would by far exceed acceptable time frames. In agreement with the editorial team, we have thus removed from the text any speculation that this band may represent the sumoylated integrase enzyme, and also cropped our Western blot gels to exclude it from the figures. Unfortunately, at this point, we do not have any explanation as to why we only observed this band in HEK293 cells, but not in Jurkat cells or in patient leukocyte samples, which may simply reflect the differential SUMO conjugation/deconjugation dynamics of different cell types.

4. Is UBA2 protein level also decreased in HIV(+) total leukocytes?

We thank the reviewer for raising this interesting point. We now went back to our patient samples, and in the 3 HIV(+) samples that we could re-assess (unfortunately, we had no more samples left from the other patients), we indeed observed a significant reduction (n=3) in UBA2 levels when compared with 3 HIV(-) controls. These results are now shown in Figure 4D, which includes a Western blot for UBA2 from a representative HIV-1-infected individual (in comparison with an uninfected control), as well a graph that represents data from all individuals analyzed. These observations led us to conclude that HIV-1 may actually target UBA2 also in vivo, which, at least in part, may contribute to sumoylation decrease in leukocytes.

5. Given that HIV infects CD4+ T cells and completes its replication cycle in approximately 24 hours, the authors should elucidate whether decreased levels of Ubc9 and global sumoylation at 48 hours post-transfection are associated with HIV life cycle.

Again, we thank the reviewer for also raising this interesting point. In fact, in both HEK293 and Jurkat cells, we do see a significant decrease in global SUMO2/3 conjugation within 24 hours of HIV-1 expression, as indicated on the graphs on Figures 1A and 1B. This suggests that HIV-1-induced modulation of cell-wide sumoylation likely begins before or as early as 24 hrs, within the replication-cycle timeframe of the virus. In our set up, because no viral particles are produced, cells survive several days post-transfection/electroporation, which allowed us to track longer-term detrimental effects of the expression of HIV-1 genes/proteins on cellular sumoylation.

Reviewer #2 (Comments to the Authors (Required)):

In this manuscript authors explored the SUMOylation events during HIV infection. They observed for the first time that SUMOylation is globally downregulated during HIV infection and that both SUMO1 and SUMO2/3 are involved, in a time dependent manner. Moreover, it seems that UBA2 expression is reduced by HIV, increasing unconjugated SUMO1. Authors showed for the first time that HIV infection induces the downregulation of SUMO pathway. The first results of the manuscript show that HIV impairs cellular SUMOylation in vitro. Data are strongly supportive since there is a significant downregulation of both SUMO1 and SUMO2/3 -conjugated proteins after 24, 48 and 72 hours.

We thank the reviewer for appreciating the novelty and importance of our findings on HIV-1-induced abrogation of cellular sumoylation. We have now extensively modified the manuscript and further enhanced the quality of our data, performed more robust statistical analyses, and presented additional supportive experimental data, where applicable.

For instance, to test whether proteasomal inhibition would prevent the virus-induced decline in cellular SUMO conjugates, we have now also treated the Jurkat cells, alongside HEK293 cells, with MG132 (new Figure 2), and included proper quantifications, statistical analyses and controls. In addition, we now also show data from Jurkat cells to demonstrate the accumulation of the free SUMO1 peptide (new Figure 3A). Furthermore, we quantified the data from new Figure 3B and provided statistical analyses. Importantly, in both HEK293 and Jurkat cells, we verified the expression of viral proteins (upon transfection or electroporation with HIV-1) by following integrase, Rev and EGFP expression in Western blot (Supplementary Figures 1A and 1B). To better understand the mechanistic basis of HIV-1-induced decline in UBA2 expression, we analyzed UBA2 mRNA levels in HIV-1-expressing cells, and assessed whether this phenomenon actually reflected the proteasomal degradation of this protein (Supplementary Figures 3A and 3B).

Please also note that in agreement with the editorial team, to ensure accuracy and correctness, we have decided to use the term "HIV-1" instead of "HIV" throughout the manuscript, including the title.

In the second figure, authors try to understand the mechanism by which SUMOylation is impaired by HIV. To this end they used the proteasome inhibitor MG132 to exclude a proteasomal degradation of SUMOylated proteins. However, they did not include the quantification of proteins of SUMOylated proteins, nor a control of the effectiveness of the treatment (Ubiquitinated proteins), please include both of them.

We thank the reviewer for pointing at this issue, which was indeed missing.

Importantly, in the revised manuscript, we now performed this experiment also in CD4+ Jurkat T-lymphocytes (Figure 2B), and corroborated the original results we had obtained in HEK293 cells.

As per the reviewer's suggestion, we now include quantifications, graphs (that represent data from several independent experiments), as well as statistical analyses. Furthermore, to verify the effectiveness of MG132 treatment, we performed Western blots for cellular ubiquitin conjugates, which indeed showed accumulation in both cell lines following MG132 exposure. These results are now shown in Supplementary Figures 2A and 2B.

They then show an increase of unconjugated SUMO1, but not of SUMO 2/3, please include it. Why did authors perform this experiment only in HEK293 cells and not also in Jurkat? Please repeat the experiments in Jurkat cells.

Again, we thank the reviewer for drawing our attention to this missing point. Importantly, we have now performed this experiment also in Jurkat cells, and once again, observed an increase in the levels of unconjugated SUMO1 peptide upon HIV-1 expression (Figure 3A). In addition, we now also include representative Western blots to show free SUMO2/3 levels in both cell lines. Please note that upon HIV-1 expression, we actually did not observe a significant accumulation of unconjugated SUMO2/3. We believe that this observation is consistent with the well-known differences in the conjugation/deconjugation behaviors of these two SUMO paralogs: in cells, the quantity of free SUMO1 is limiting because most of this peptide is found attached to certain substrate proteins, such as RanGAP1 and PML. On the other hand, free SUMO2/3 is abundantly expressed, however, most of it remains unconjugated unless cells are exposed to stress conditions. This difference in the conjugation behavior actually renders SUMO1 much more sensitive and responsive to changes in the enzymatic machinery, and also considering that the baseline abundance of free SUMO2/3 relative to its conjugated form is quite elevated to begin with, could explain why HIV-1-driven UBA2 loss did not result in a noticeable accumulation of free SUMO2/3. We have now included a statement to discuss this point in the text.

The last experiments of the figure show that UBA2 is downregulated by HIV infection, while SAE1 and Ubc9 are not affected and these are strongly supportive data.

We thank the reviewer for appreciating the quality and relevance of this important piece of data.

Authors showed that global leukocytes SUMOylation is impaired independently of HIV-induced cytopenia during HIV disease. To do this they first checked SUMO conjugation levels in HIV+ patients, that seems to confirm the data obtained in vitro and they are strongly supportive. Moreover, they showed that depletion of CD4+ cells from peripheral blood of HIV- individuals has not any effect on SUMO conjugation. The last experiment shows that

antiretroviral therapy (ART) is able to rescue the downregulation of SUMO pathway induced by HIV infection and data support the hypothesis. All these last data are strongly supportive.

Again, we thank the reviewer for this accurate summary of our findings, and also for appreciating the quality, importance and relevance of these experiments/data.

I have some points in the text that should be addressed:

Authors show in figure 1A, in a SUMO1 western blot a band they claim to be a SUMO1-modified HIV-1 integrase. How do they explain the absence of this band in the same experiment performed in Jurkat cells?

Because this heavily sumoylated band appears only in HIV-1-infected cells and also because its size corresponds to the size of SUMO1-conjugated HIV-1 integrase, we initially thought and speculated that it could possibly be the sumoylated form of this enzyme. Unfortunately, at this point, we do not have an explanation as to why we only observed this band in HEK293 cells, but not in Jurkat cells or in patient leukocyte samples, which may simply reflect the differential SUMO conjugation/deconjugation dynamics of different cell types. It would be interesting to determine the identity of this band; however, upon further consultation with the editorial team, we have come to an agreement that further studies along this line may be beyond the scope and purpose of this study, and would by far exceed acceptable time frames. In agreement with the editorial team, we have thus removed from the text any speculation that this band may represent the sumoylated integrase enzyme, and also cropped our Western blot gels to exclude it from the figures.

ART acronym is used in the text but not mentioned, please explain it.

We thank the reviewer for drawing our attention to this missing point. We have now explained this acronym (ART: anti-retroviral therapy) in the text, on page 10.

In the discussion (page 9), authors mention the effects of viruses on SUMO pathway. It would be complete to also add an older and important finding 10.1016/j.molcel.2004.11.007 Boggio et al Mol Cell 2004

We thank the reviewer for reminding us of this important work from Susanna Chiocca, Ron Hay and the others. We have now cited this paper in the discussion, and included the sentence "The avian adenovirus CELO encodes a protein called Gam1, which inactivates the SUMO E1 heterodimer and reduces the stabilities of the SUMO E1 and E2 (UBC9) enzymes, thereby efficiently downregulating host cell sumoylation during infection (Boggio et al.)" on page 13.

In material and methods authors explain the aim of the experiment. Is it the right section for this?

We agree with the reviewer, and therefore removed this short paragraph from the Materials and Methods section in the revised manuscript.

Minor grammar and spelling errors are present, please check.

We thank the reviewer for raising our awareness on this issue. We have indeed found some grammatical and spelling errors, and did our best to correct them in the revised manuscript.

Why authors split blot lanes, in particular Figure 3C? Is it possible to show the original blots?

Actually, these patient samples were run on the same protein gel, which also contained some other samples in between. In order to show only the relevant ones, we spliced the lanes in the final image. As per the reviewer's request, below we show the original blot, where we also marked the spliced lanes that were used to construct the final image. Unfortunately, at this time, we have no more patient samples left to re-run and present a better, unspliced image. Please note that in the revised manuscript, this is now Figure 4C.

Reviewer #3 (Comments to the Authors (Required)):

In this paper the authors mainly observed that HIV transfected plasmids expression on 293FT and Jurkat cells induce a decrease of sumoylation with a specific decline of UBA2. This study also demonstrates a significant decrease in total leukocyte sumoylation profile from HIV-induced cytopenia, absent from patients under ART treatment. The results seems really promising and the understanding of the mechanism should be a plus for this story, but it

seems that it's not the main objective for this publication.

We thank the reviewer for appreciating the novelty, promise and importance of our findings on HIV-1-induced abrogation of cellular sumoylation. We have now extensively modified the manuscript and further enhanced the quality of our data, performed more robust statistical analyses, and presented additional supportive experimental data, where applicable. For instance, to test whether proteasomal inhibition would prevent the virus-induced decline in cellular SUMO conjugates, we have now also treated the Jurkat cells, alongside HEK293 cells, with MG132 (new Figure 2), and included proper quantifications, statistical analyses and controls. In addition, we now also show data from Jurkat cells to demonstrate the accumulation of the free SUMO1 peptide (new Figure 3A). Furthermore, we quantified the data from new Figure 3B and provided statistical analyses. Importantly, in both HEK293 and Jurkat cells, we verified the expression of viral proteins (upon transfection or electroporation with HIV-1) by following integrase, Rev and EGFP expression in Western blot (Supplementary Figures 1A and 1B).

We agree with the reviewer that understanding the mechanisms underlying the HIV-1-induced UBA2 loss (and downregulation of cellular sumoylation) would be very valuable, and we hope that our study will pave the way for future endeavors that will address this question. Along these lines, we have actually performed some additional experiments in the revised manuscript. Firstly, in an effort to understand the basis of UBA2 loss in HIV-1-expressing cells, we now performed real-time PCR analyses to assess UBA2 mRNA levels, however, we did not detect significant differences in the transcription of this gene upon transfection of HEK293 cells or electroporation of Jurkat cells with HIV-1 (Supplementary Figure 3A). We have also established that proteasome inhibition (by MG132) did not stabilize UBA2 in HIV-1-expressing cells (Supplementary Figure 3B). Therefore, we conclude that the virus actually induces non-proteasomal proteolytic degradation of this enzyme.

Please also note that in agreement with the editorial team, to ensure accuracy and correctness, we have decided to use the term "HIV-1" instead of "HIV" throughout the manuscript, including the title.

Major points:

Figure 1/S1 and Main text:

Figure S1 show GFP expression on HEK293 and demonstrated that the HIV plasmid express normally the HIV proteins. It should be great to precise the time point of this IF acquisition.

This immunofluorescence image (Supplementary Figure 1 in the original manuscript) was acquired 24 hours post-transfection of HEK293 cells. However, in the revised manuscript, in order to verify the transfection efficiency of HEK293 cells, or the electroporation efficiency of Jurkat cells, and also to confirm viral protein expression, we performed Western blot analyses in both cell lines. These analyses (results shown in

Supplementary Figures 1A and 1B) confirmed that EGFP, HIV-1 integrase and HIV-1 Rev proteins were correctly expressed in HEK293 and Jurkat cells (at all time points checked: 24, 48 and 72 hours). In the revised manuscript, we now replaced the old immunofluorescence image with these new, better and more comprehensive data.

Due to the fact, the authors want to demonstrate a correlation of the decrease of the sumoylation profile (SUMO1 and SUMO2) and the expression and presence of HIV proteins, it should be great to see the GFP profile and another viral protein (as CA expression) on the WB.

We thank the reviewer for raising this important point. As mentioned above, we now include a Western blot that shows EGFP, HIV-1 integrase and HIV-1 Rev expression in both HEK293 and Jurkat cells at the relevant time points (24, 48 and 72 hours) (Supplementary Figures 1A and 1B).

The data results from the transfection of an HIV-plasmid on HEK293 or Jurkat cells. In order to be sure that it's not just the over-expression of some proteins that can induce the decline of the sumoylation profile, it should be great to have an internal control with another plasmid that express only GFP or another protein.

Again, we thank the reviewer for drawing our attention to this important point. We now include this control for both HEK293 and Jurkat cells in Supplementary Figure 1C. Importantly, we now confirm that overexpression of EGFP alone indeed does not cause a reduction in global cellular sumoylation (nor does it alter the overall sumoylation profile).

It should be also interesting to reproduce this data not in the context of transfected cells with a plasmid but with the transduction/infection of the cell lines with the same virus (produced and harvested from HEK293 but pseudotyped with VSV-G for example).

We agree with the reviewer that the proposed experiment with a transduction/infection set-up would be indeed very valuable and more physiologically relevant. However, setting up such a system that would make it possible to infect cells with live HIV-1 viral particles would require specialized facilities with proper biosafety measures, which we unfortunately lack at our institutions. Therefore, following discussions with the editorial team and upon their approval, we continued our studies with the transfection/electroporation method, and would like to ask for the reviewer's understanding in this matter.

Please note that in this current set up that we use, the HIV-1 lentivirus that we transfect into HEK293 cells or electroporate into CD4+ Jurkat T-lymphocytes is still capable of reverse transcription, integration and replication, as it carries and expresses all of the structural, accessory and regulatory HIV-1 genes: gag, pol, tat, rev, vif, vpr, vpu, nef and env (an EGFP cassette was inserted within env). Therefore, this

system can exquisitely simulate viral infection, making it possible to investigate and document potentially detrimental and subversive effects of viral proteins on host cell mechanisms, such as sumoylation.

From this perspective, our study is a first to demonstrate that the introduction of HIV-1 into cells, allowing replication and viral protein expression, can antagonize this critical post-translational modification. We hope that this work will pave the way for further future studies and prompt other groups (with access to proper biosafety measures) to further investigate HIV-1 / host sumoylation interactions in more natural set ups.

This observation is still observable with an HIV-2 plasmids/viruses?

This is indeed a very interesting question, but to answer it would require a whole set of experiments to be performed on a different virus, which would exceed acceptable time frames. Upon further discussions with the editor, we have come to an agreement that this point, though very relevant and interesting, may be beyond the context of the current study. However, we would very much be interested in investigating this question in a future study.

The authors do the hypothesis that the bands at around 50 to 60 kDa indicated by an arrow (Fig 1A) should be the integrase. I think is pure speculation here. We know that the integrase should be SUMOylate in order to optimize the nuclear localization and the function, but the authors need to proof at least by an anti-Integrase and to observe or not if the WB size are the same. Strangely, during the kinetic we didn't observed any increase of expression of this band (always same intensity @24-48 and 72h post transfection). Moreover, we didn't observed this band on Jurkat cells WB and Fig2A with MG132 treatment exactly the same bands appears on the control line.

Because this heavily sumoylated band appears only in HIV-1-infected cells and also because its size corresponds to the size of SUMO1-conjugated HIV-1 integrase, we initially thought and speculated that it could possibly be the sumoylated form of this enzyme. Unfortunately, at this point, we do not have an explanation as to why we only observed this band in HEK293 cells, but not in Jurkat cells or in patient leukocyte samples, which may simply reflect the differential SUMO conjugation/deconjugation dynamics of different cell types. It would be interesting to determine the identity of this band; however, upon further consultation with the editorial team, we have come to an agreement that further studies along this line may be beyond the scope and purpose of this study, and would by far exceed acceptable time frames. In agreement with the editorial team, we have thus removed from the text any speculation that this band may represent the sumoylated integrase enzyme, and also cropped our Western blot gels to exclude it from the figures.

It should be a plus to better describe how the abundance profile of SUMO1 and SUMO2/3 is calculated. Is it a quantification performed on a specific

bands/molecular weight or on all the profile line?

In all SUMO Western blots, we have actually quantified the entire smear pattern representing all of the SUMO conjugated proteins, especially the high molecular weight ones. This includes, in former Figures 1A and 2A, anything above the presumed “integrase” band, which we have now removed from the revised images. As an example, the images below indicate which parts of the blots (within the red frames) were quantified in Figure 4A.

Figure 2 and Main text:

Fig2C, it should be more significant to performed and add the quantification of the abundance of UBA2, SAE1 and UBC9 during the kinetic of expression, and also performed to the statistics. We can see that the data are strong but the quantifications. GFP or Capsid expression on the WB should be a plus (but minor here).

We thank the reviewer for drawing our attention to this missing point. This experiment was indeed performed several times with consistent results in each time. We have now performed quantifications, constructed graphs that represent data from ≥ 3 independent experiments, and performed statistical analyses. In the revised manuscript, these results are now shown in Figure 3B.

Figure 3 and Main text:

It should be interesting to investigate the degradation/decline of UB2A, and also of SAE1 and UBC9 (due to the fact that SUMO2/3 is more impacted than SUMO1, the phenotype of decline if existing can be different).

We thank the reviewer for raising this interesting point. We now went back to our patient samples, and in the 3 HIV(+) samples that we could re-assess (unfortunately, we had no more samples left from the other patients), we indeed observed a significant reduction ($n=3$) in UBA2 levels when compared with 3 HIV(-) controls. These results are now shown in Figure 4D, which includes a Western blot for UBA2 from a

representative HIV-1-infected individual (in comparison with an uninfected control), as well a graph that represents data from all individuals analyzed. These observations led us to conclude that HIV-1 may actually target UBA2 also in vivo, which, at least in part, may contribute to sumoylation decrease in leukocytes. Unfortunately, we did not have enough sample left to assess SAE1 or UBC9 levels.

The authors claimed that the decline of sumoylation observed on SUMO1 and more SUMO2/3 potentially came from HIV-induced cytopenia, absent from patients under ART treatment. Is it possible to put in culture for 24h or 48h PBMCs depleted or not of CD4 cells (healthy donor) and observed or not if the sumoylation profile is changed?

This is an interesting point, and following the reviewer's suggestion, we actually did try to culture total leukocytes (before and after CD4+ cell depletion) from a healthy donor in order to analyze their sumoylation profiles by Western blot. Unfortunately, despite several trials, all our attempts to keep these cells in culture for up to 48 hours have failed. Given the time and consumables invested in these trials, we ask for the reviewer's understanding in this matter.

In order to demonstrate that the sumoylation phenotype from total leukocyte came from mainly of the non CD4+ cells, the authors performed a depletion of CD4+ cells from PBMCs. The WB from Fig.3B, show we absence of difference between the total and CD4+ populations. I understand that the CD4+ cells represent only around 14% of the leukocytes, but it should be great to show the sumoylation profile of the CD4+ purified cells vs. total and CD4+ populations.

We now show the sumoylation profile of CD4+ fraction (in comparison with that of total leukocytes, before CD4+ cell depletion) in Supplementary Figure 4B. Please note that because CD4+ cells constitute a tiny fraction of total leukocytes (8-15%), the level of actin is also considerably low in this fraction. However, this Western blot once again supports our findings that the contribution of the CD4+ compartment to total leukocyte sumoylation is considerably low, if not negligible. Also please note that despite our multiple attempts, we could not obtain an informative SUMO2/3 Western blot result from these CD4+ samples, thus we chose to show only the SUMO1 blot.

Discussion:

it's writing "HIV is also known to interact with a subset of non-CD4+ cells, including macrophages and others [1]." I hope it's just a little issue of writing, but macrophage are CD4+ cells, like dendritic cells and can be infected by HIV, sure less compare to T CD4+ cells.

We thank the reviewer for drawing our attention to this mistake. This was indeed an oversight on our part, and we have now corrected the sentence in the revised manuscript, which now reads "HIV-1 is also known to interact with a subset of non-CD4+ cells (Cohen, 2011)".

Minor points:

Figure 1/S1 and Main text:

The HIV plasmid used for this experiment is pNL43-dEGFP, it should be great to really precise that all the HIV proteins were present unless Env (GFP in place). With this information we know that all accessory proteins (Vpr, Vif, Vpu ...) are present and some of them should participate to this decline of sumoylation. In order to investigate the mechanism (future studies I hope) should be a plus to have pNL43-dEGFP deleted from Vpr, Vpu, Vif or to over-express only HIV-1 and/or HIV-2 accessory proteins (Vpr, Vpu, Vif, Nef, Vpx ...).

Once again, we thank the reviewer for raising this very important issue. As mentioned earlier, the HIV-1 lentivirus that we use carries and expresses all of the structural, accessory and regulatory HIV-1 genes: gag, pol, tat, rev, vif, vpr, vpu, nef, except that an EGFP cassette is present within env. We have now emphasized this point in the manuscript text.

We also agree with the reviewer that our work will hopefully pave the way for future endeavors to reveal the contribution of individual HIV-1 proteins (or its regulatory, structural or accessory genes) to sumoylation decrease. In the meantime, in an effort to elucidate which viral genes may play role in the downregulation of cellular sumoylation, we have actually carried out a preliminary analysis (n=2) and transfected HEK293 cells with either the HIV-1 structural gag and pol genes, or the HIV-1 regulatory rev gene, or gag/pol and rev altogether (these were the only viral genes that were available to us for further study). Interestingly, we found that upon overexpression of gag/pol and rev together (but neither of gag/pol alone, nor of rev alone), cellular sumoylation was downregulated - in the absence of other regulatory (tat) and accessory (vif, vpr, vpu and nef) elements. Consistent with the literature, Rev was crucial for the synthesis of viral proteins, such as the integrase enzyme (please refer to the image below). These data suggest that one of the proteins processed from the Gag or Pol polypeptides (by the help of Rev) might be the culprit responsible for attacking host sumoylation (or might at least play some role in this process). Here, we share these preliminary results, which were not included in the revised manuscript, with the reviewer. In the representative experiment shown below, HEK293 cells were transfected either with the HIV-1 lentivirus (as a positive control, labeled as HIV), or with vectors encoding the indicated viral genes. Global SUMO2/3 profile was assessed by Western blot 48 hours post-transfection. 1 µg of total DNA was used for transfections, except in the last lane where the DNA amount was doubled.

FigS1, that is the profile on Hela cells of SUMO1 expression after HIV expression? How many time this experiment was performed (n=?)?

This experiment was performed 3 times, with consistent and similar results in each time. In the revised manuscript, we now include a graph to represent the data from these independent experiments, along with statistical analyses (new Supplementary Figure 1D). We thank the reviewer for prompting us to improve this figure. We now also show the expression levels of EGFP, HIV-1 Rev and HIV-1 integrase proteins for this representative experiment.

Figure 2 and Main text:

The authors observed more SUMO1 (unconjugated) after HIV expression Fig.2B. Is it the case for SUMO 2/3?

We thank the reviewer for drawing our attention to this missing point. First of all, we have now performed this experiment also in Jurkat cells, and once again, observed an increase in the levels of unconjugated SUMO1 peptide upon HIV-1 expression (Figure 3A). In addition, we now also include representative Western blots to show free SUMO2/3 levels in both cell lines. Please note that upon HIV-1 expression, we actually did not observe a significant accumulation of unconjugated SUMO2/3. We believe that this observation is consistent with the well-known differences in the conjugation/deconjugation behaviors of these two SUMO paralogs: in cells, the quantity of free SUMO1 is limiting because most of this peptide is found attached to certain substrate proteins, such as RanGAP1 and PML. On the other hand, free SUMO2/3 is abundantly expressed, however, most of it remains unconjugated unless cells are exposed to stress conditions. This difference in the conjugation behavior actually renders SUMO1 much more sensitive and responsive to changes in the

enzymatic machinery, and also considering that the baseline abundance of free SUMO2/3 relative to its conjugated form is quite elevated to begin with, could explain why HIV-1-driven UBA2 loss did not result in a noticeable accumulation of free SUMO2/3. We have now included a statement to discuss this point in the text.

Figure 3 and Main text:

Fig 3A, have you try to see if you observed an increase of SUMO1 (unconjugated) on the WB profile (minor)?

This is also a very interesting point, unfortunately, despite all our attempts and for the reasons we cannot explain, we were unable to detect the free SUMO1 peptide on Western blots produced from the patient samples.

Discussion:

it's writing "Of note, we have seen in vitro that HIV efficiently reduces sumoylation by both SUMO1 and SUMO2/3 needed to explore this."

The authors just need to comment/add that for the total leukocyte the sumoylation profile from HIV patient is more impacted on SUMO2/3 than SUMO1. This difference can be easily explain by the difference between immune primary cells and cancer cell lines (HEK293 and Jurkat) that have evolute to cancelled and/or more activate some specific pathway.

We thank the reviewer for mentioning this important and critical point. We have now included the sentence "However, we cannot rule out the possibility that primary and transformed cells may behave differently pertaining to the cellular pathways and mechanisms that are activated or silenced upon viral infection; for instance, paralog-specific suppression of sumoylation" in the Discussion on Page 14.

it's writing "Alternatively, one of the HIV proteins may have a previously uncharacterized STUbL activity or may interact and activate a host STUbL to initiate proteasome-dependent degradation of sumoylated proteins." The authors should just mentioned that some HIV accessory proteins are specialize to link the E3 ligase complex like Vpr, Vpx in order to induce some proteins degradation.

Again, we thank the reviewer for this important reminder. We have now added the sentence "Importantly, the viral accessory proteins Vif, Vpx (found in HIV-2) and Vpu are known to counteract host restriction factors by serving as adaptors between these factors and certain ubiquitin E3 ligases and inducing their degradation (Seissler, 2017)" in the Discussion, on Page 15.

it's writing "A recent study revealed that Klebsiella pneumoniae reduces sumoylation to restrict host defense responses by either increasing the levels of SENP2, a SUMO- deconjugating enzyme, or by interfering with let-7 family of microRNAs that target the SUMO transcripts [43, 58]." Another paper (2014) also demonstrate that interferon controls SUMO availability via the Lin28 and

let-7 axis to impede virus replication (DOI: 10.1038/ncomms5187).

The reference number 43 which we had indeed cited in this sentence was actually the paper the reviewer mentions (Sahin et al, 2014, Nat Commun, interferon controls SUMO availability via the Lin28 and let-7 axis to impede virus replication).

February 4, 2022

RE: Life Science Alliance Manuscript #LSA-2021-01103R

Prof. Umut Sahin
Bogazici University
Department of Molecular Biology and Genetics
Center for Life Sciences and Technologies
Bogazici University
Istanbul 34342
Turkey

Dear Dr. Sahin,

Thank you for submitting your revised manuscript entitled "Human Immunodeficiency Virus Type 1 Impairs Sumoylation". We would be happy to publish your paper in Life Science Alliance pending final revisions necessary to meet our formatting guidelines.

- please address the remaining comments from Rev 1 and 3, excluding Rev 1 point 2
- please upload your Table in editable .doc or excel format
- please add ORCID ID for secondary corresponding author-they should have received instructions on how to do so
- please use the [10 author names, et al.] format in your references (i.e. limit the author names to the first 10)

A. FINAL FILES:

B. MANUSCRIPT ORGANIZATION AND FORMATTING:

**Submission of a paper that does not conform to Life Science Alliance guidelines will delay the acceptance of your

manuscript.**

The license to publish form must be signed before your manuscript can be sent to production. A link to the electronic license to publish form will be sent to the corresponding author only. Please take a moment to check your funder requirements.

Sincerely,

Reviewer #1 (Comments to the Authors (Required)):

While the authors have deleted the controversial issue regarding integrase sumoylation, the current revised manuscript did have some additional improvements.

1. The explanation regarding free SUMO-1 but not free SUMO-2/3 in HIV-1 infected cells was not satisfied. One alternative explanation is that the HIV-1 infection altered SUMO-1 but not SUMO-2/3 gene expression.
2. Similarly, the HIV-1 infection-caused reduction of UBA2 protein could be due to the infection decreased gene expression of UBA2 given that the protein reduction was observed after 48 h. I strongly suggest the authors should explore some mechanistic studies instead of using MG132 treatment alone.

Reviewer #2 (Comments to the Authors (Required)):

Authors have significantly improved their work. Hence the manuscript can now be accepted for publication.

Reviewer #3 (Comments to the Authors (Required)):

The authors have largely answered my major and minor questions and criticisms/points. The new version is much more relevant from the point of view of results and discussion. The authors still demonstrated and explored the SUMOylation events during HIV infection. For the first time, they demonstrated that SUMOylation is downregulated during HIV infection and that both SUMO1 and SUMO2/3 are involved. For some more complex questions and not directly within the scope of this paper, the authors seem to have contacted the editors to proceed with figure modifications or others.

However, I would have two small corrections to add:

- 1- page 10 - add ")" after Table10) because it is missing, like (.....(Table 10)).
- 2- in order to improve the reading of figures 2 and above S2. It would be nice to integrate directly into the figure that the MG132 is added during the last 24 hours before harvesting and cell lysis. This information is present in the legends, but looking at the figure we have the impression that the MG132 is always present since the beginning of the experiment. Example after HEK293 / Jurkat (+MG132), put something like HEK293 / Jurkat (+MG132, added last 24h).

February 8, 2022

RE: Life Science Alliance Manuscript #LSA-2021-01103RR

Prof. Umut Sahin
Bogazici University
Department of Molecular Biology and Genetics
Center for Life Sciences and Technologies
Bogazici University
Istanbul 34342
Turkey

Dear Dr. Sahin,

Thank you for submitting your Research Article entitled "Human Immunodeficiency Virus Type 1 Impairs Sumoylation". It is a pleasure to let you know that your manuscript is now accepted for publication in Life Science Alliance. Congratulations on this interesting work.

DISTRIBUTION OF MATERIALS:

Again, congratulations on a very nice paper. I hope you found the review process to be constructive and are pleased with how the manuscript was handled editorially. We look forward to future exciting submissions from your lab.

Sincerely,
